# A Multi-Correlation Peak Phase Deblurring Algorithm for BeiDou B1C Signals in Urban Environments

Xu Yang [1], Wenquan Feng [1], Chen Zhuang [2,*], Qiang Wang [1], Xu Yang [1] and Zhe Yang [1]

1    School of Electronic & Information Engineering, Beihang University, Beijing 100080, China;
     by1702151@buaa.edu.cn (X.Y.)
2    Hefei Innovation Research Institute of Beihang University, Hefei 230012, China
*    Correspondence: zhuangchen0214@buaa.edu.cn

**Abstract:** With the widespread global application of BeiDou navigation, BeiDou B1C signaling based on Quadrature Multiplexed Binary Offset Carrier (QMBOC) modulation is expected to be extensively used in urban environments due to its wider signal bandwidth, smaller code pseudorange measurement errors, and stronger multipath capabilities. Despite offering higher positioning accuracy and secondary modulation characteristics of the BeiDou, B1C signals introduce the challenge of multiple peaks in the autocorrelation function. This leads to phase ambiguity during signal acquisition and tracking, resulting in positioning deviations of tens or even hundreds of meters. In urban environments, such deviations give rise to significant practical application issues. To address this problem, we have designed a multi-loop structure for the synchronous tracking of B1C signals and proposed a multi-peak phase-deblurring algorithm specifically tailored for the BeiDou B1C signal in urban environments. This algorithm considers the coupling relationship between the code and the carrier loops, and by matching the structural design of multiple loops, it achieves a precise and unambiguous phase estimation of the pseudocode, enabling the stable tracking of the entire loop for the BeiDou B1C signal. Simulation and actual testing demonstrate that the algorithm exhibits an error less than 0.03 for chip intervals when the signal-to-noise ratio is greater than −20 dB. Additionally, the accuracy can be improved by adjusting the set conditions, making it suitable for urban environments.

**Keywords:** BeiDou; B1C; QMBOC; urban environment; signal model; tracking loop; phase deblurring algorithm

## 1. Introduction

Urban environments are important scenarios for Global Navigation Satellite System (GNSS) applications, especially with the rapid development of technologies such as autonomous driving, intelligent transportation, and smart ports in recent years. These technologies impose higher requirements on the positioning accuracy and reliability of GNSS, demanding higher performance and robustness in the signal processing and information processing of navigation receivers. In terms of signal processing, this requires higher precision and stability in the signal synchronization process (tracking loop) to adapt to various complex application scenarios in urban environments. The third-generation BeiDou satellite navigation system, an independently developed GNSS system in China, includes the newly added B1C civil signal. This new signal, based on QMBOC modulation, has a wider signal bandwidth, smaller code pseudorange measurement errors, and stronger multipath mitigation capabilities [1]. It is expected to find extensive applications in urban environments. However, its spreading and secondary modulation characteristics introduce the problem of multiple peaks in the autocorrelation function, causing phase ambiguity in the acquisition and tracking of B1C signals. Incorrect signal tracking can result in positioning errors of tens or even hundreds of meters, leading to significant application issues in urban environments. Therefore, non-ambiguity tracking techniques for B1C signals have gained increasing attention in the academic community.

In the realm of modern GNSS, distinctive signals such as the BOC signal families (CBOC, MBOC, AltBOC, QMBOC) exhibit distinct advantages across various application scenarios. They possess the capacity to meet compatibility and interoperability requirements among global navigation satellite systems while simultaneously delivering heightened positioning precision and availability. Currently, the four major satellite navigation systems worldwide, namely the GPS from the United States, the BDS from China, the Galileo from the European Union, and the Glonass from Russia, widely adopt the Binary Offset Carrier (BOC) modulation and its derivative forms. These modulation techniques, such as Composite BOC (CBOC), Multiplexed BOC (MBOC), and Alternative BOC (AltBOC), have become the primary modulation methods for the next generation of satellite navigation systems [2]. Furthermore, with the further development of satellite navigation systems and the continuous addition of satellite navigation systems from various countries, it is inevitable that more new BOCs and their derivative modulation signals will be adopted. The research conducted by Foucras et al. highlights the potential benefits of BOC modulation for GNSS. The higher positioning accuracy and improved availability provided by BOC modulation make it a favorable choice for meeting the demanding requirements of modern navigation applications. Moreover, the compatibility and interoperability offered by BOC modulation ensure seamless integration and collaboration among different satellite navigation systems worldwide [3]. Kao et al. conducted an analysis of the role of MBOC modulation, which is employed by modern global navigation satellite systems, as well as the issue of ambiguity caused by multiple peaks in the autocorrelation function. They proposed a class of unambiguous code tracking techniques to address this problem. The results demonstrated that the proposed method is simple to implement, free from ambiguity, and achieves acceptable performance even in the presence of multipath and noise [4]. Harris et al. conducted a study to analyze and evaluate the anti-interference performance of BOC modulation in multiple Global Navigation Satellite Systems by modeling the impact of multipath interference on signal tracking. The research addressed the condition whereby M-code receivers exhibit more multipath errors compared to P(Y)-code receivers and demonstrated the consistency of the zero-mean condition in carrier phase measurements. Conversely, pseudorange measurements exhibited a zero-mean condition only under certain tracking conditions [5]. Rusu-Casandra et al. presented a novel methodology aimed at effectively mitigating the adverse effects of Continuous Wave Interferences (CWIs) on the MBOC Galileo navigation signal. The researchers investigated various filtering techniques as potential solutions to counteract the influence of CWIs on the signal [6]. Zhang et al. proposed an extended AltBOC signal that utilizes the phase rotating technique to achieve unbalanced power distribution between the upper and lower sidebands. This innovative modulation approach provides increased flexibility in power control compared to conventional Dual-Quadrature Phase Shift Keying (DualQPSK) and AltBOC signals. The researchers present analytical implementations that demonstrate the optimality of the proposed signal, thereby maximizing power efficiency [7]. Yan et al. addressed the limited flexibility in power allocation of BOC modulation signals by proposing a universal AltBOC modulation scheme. This novel approach allows for adjustable power distribution ratios according to specific requirements while maintaining the same functionality as AltBOC [8].

The BeiDou B1C signal employs BOC modulation, addressing limited bandwidth constraints in satellite navigation systems for improved positioning accuracy, interference resistance, and multipath mitigation. Nevertheless, BOC signals exhibit multi-peak autocorrelation characteristics, causing synchronization ambiguity. Ambiguity resolution is vital for effective BOC signal utilization and is a key challenge in new-generation GNSS signal research. Various techniques aim to enhance signal acquisition, tracking, and positioning, ultimately unlocking the benefits of BOC modulation and advancing GNSS signal research for robust navigation systems. Yang et al. proposed a novel receiver architecture that performs correlation on a variable Intermediate Frequency (IF). By replicating the code chip and up-converting it to an appropriate IF, joint code and carrier correlation are effectively carried out, resulting in correlation peaks as sharp as those permitted by the upper IF limit.

By gradually changing the IF, not only can the correlation peaks be strengthened, but it also provides a method that allows for reliable and unambiguous locking onto the main peak without generating ambiguities or erroneous lockings [9]. Attia et al. proposed an effective solution, based on the Autocorrelation Function (ACF), to eliminate side peaks associated with the integer modulation order in Binary Offset Carrier (m,n) (BOC(m,n)) signals. This solution demonstrates excellent performance for mitigating multipath and noise effects and has been employed in the Global Positioning System (GPS) and the Russian Global Navigation Satellite System (GLONASS) [10]. Anantharamu et al. discussed the advantages of pre-filtering techniques and proposed a pre-filtering technique based on the zero-forcing and minimum mean square error equalization concepts. The BOC subcarrier is modeled as a filter that introduces secondary peaks in the autocorrelation function, enabling unambiguous tracking and allowing for the shaping of autocorrelation [1]. Deng et al. proposed a method based on multiple correlators and linear fitting to achieve unambiguous tracking. In their study, the authors addressed the challenge of unambiguous tracking in satellite navigation systems. They introduced a novel approach that utilizes multiple correlators in combination with linear fitting techniques. This method aims to overcome the ambiguity issue associated with tracking signals affected by multipath interference and thermal noise. Simulation results demonstrated that the proposed method performs well in the presence of multipath interference and thermal noise [11]. Sun et al. proposed an unambiguous synchronization method based on reconstructed correlation functions to address the issue of ambiguity. Simulation results demonstrate that the method completely eliminates the threat of side peaks and significantly reduces ambiguity in the synchronization process of BOC signals. Furthermore, the method exhibits superior performance in multipath mitigation compared to the ACF of BOC signals [12]. Chengtao et al. modified the Dual Phase Estimator (DPE) algorithm by introducing frequency-hopping waveforms in the prompt signal correlation process of the subcarrier tracking loop. They provided two different reference frequency-hopping waveforms and their corresponding receiver structures. These modifications effectively enhance the unambiguous tracking and multipath resistance capabilities of BOC signals [13].

Designing receiver algorithms for practical BOC signals involves selecting unambiguous synchronization methods based on modulation characteristics, order, and desired reception performance. Recent research has concentrated on developing such algorithms, specifically tailored to BeiDou B1C signals. This focus is crucial in urban settings marked by a complex infrastructure, where multipath and signal blockage prevail. BeiDou B1C signals excel in urban scenarios due to their wider bandwidth, reducing code pseudorange errors and enhancing multipath resilience. This broader bandwidth conveys more information, improving positioning accuracy and robustness. Reduced code pseudorange errors boost precision and reduce multipath interference, enhancing overall system performance. The signal's enhanced multipath resistance further fortifies its urban adaptability. Research on unambiguous acquisition and tracking algorithms for BeiDou B1C signals addresses the need for reliable positioning in challenging urban conditions. Researchers leverage the signal's superior performance to develop robust receiver algorithms and positioning solutions, mitigating multipath and signal blockage challenges in urban environments.

Guo et al. presented a "Data + Pilot" joint tracking strategy in cycle tracking to improve the accuracy of demodulating navigation information based on the phase relationship between the data channel and the pilot channel. They proposed three different complexity levels of joint methods, including "Correlator Output Amplitude Summation", "Discriminator Output Linear Combination", and "Cycle Filter Output Linear Combination [14]". Gao et al. conducted an analysis of the ranging potential and tracking challenges of the Single-Sideband Complex Binary Offset Carrier (SCBOC) modulation signal. They proposed an exemplary scheme to explore the ultra-high-precision ranging potential of the BeiDou B1 signal [15]. Hao et al. proposed a combined open-loop and closed-loop tracking method for the BeiDou B1C signal to address the issues of low accuracy in open-loop tracking and long acquisition time in closed-loop tracking [16]. The research conducted

by Hao et al. contributes to the advancement of signal tracking techniques, specifically for the BeiDou B1C signal. Their focus on locally non-matched coherent functions and the utilization of the Adaptive Subspace Power Estimation and Cancellation Technique (ASPeCT) algorithm showcases their efforts in enhancing signal tracking performance. The theoretical analysis, algorithm optimization, and simulation analyses provide valuable insights into the algorithm's behavior and potential improvements, ultimately aiming to enhance the accuracy and robustness of BeiDou B1C signal tracking [17–19]. Sun et al. constructed novel local correlation signals and proposed methods that significantly mitigate ambiguities in code tracking. Their approach involved utilizing techniques such as sub-cross-correlation shift [20], linear joint tracking algorithms [21], an unambiguous Cross-Assisted Tracking (CAT) loop [22], and the dual-frequency flashing tracking loop structure [23]. These proposed techniques demonstrated superior performance in terms of main lobe width, the phase resolution curve, and multipath resistance. Some studies utilize additional information to assist in the acquisition and tracking of B1C signals. For instance, Wu et al. proposed a B1C/B2a joint tracking architecture for dual-frequency BeiDou receivers based on adaptive Kalman filtering and extended integration time [24]. Wang et al. established a linearized mathematical model for cross-frequency Doppler-assisted carrier phase tracking loops. They derived the calculation equation for thermal noise-induced jitter in the auxiliary tracking loop and investigated its dynamic stress response characteristics [25]. Wu et al., on the other hand, improved the tracking sensitivity and robustness of B1C signals by employing unambiguous tracking techniques based on pseudo-correlation functions. They introduced a Doppler-assisted B1C/B2a joint tracking loop [26].

In summary, the main contributions of this paper are as follows:

1.  This paper constructs a mathematical model and structure for the BeiDou B1C signal, exploring autocorrelation properties and power spectrum density characteristics, and it addresses ambiguity in BOC signals. This foundational research informs the design and validation of the tracking loop synchronization algorithm.
2.  This study introduces a custom multi-loop structure for synchronized B1C signal tracking, along with a specialized multi-peak phase deblurring algorithm tailored to urban BeiDou B1C signals. This coordinated design ensures accurate pseudocode phase estimation and stable tracking.
3.  This study simulated tracking algorithm performance in urban environments at $-20\,\text{dB}$ SNR, assessing it across four dimensions: coherent integration, non-coherent integration, Doppler frequency error, and code phase estimation. Subcarrier tracking was also analyzed.

## 2. Proposed Method

As shown in Figure 1, the research logic and analysis paradigm for the BeiDou B1C signal and synchronization loop can be summarized based on the aforementioned studies. This paper addresses the various challenges faced by the application of BeiDou B1C signals in urban environments. Aiming to tackle these challenges, this paper proposes a multi-loop structure suitable for B1C signal synchronization and tracking, and based on this structure, it presents a multi-peak phase deblurring algorithm specifically designed for BeiDou B1C signals in urban environments. This algorithm considers the coupling relationship between code and carrier loops and achieves an accurate and unambiguous phase estimation of a pseudocode by matching the structure design of multiple loops, the enabling stable tracking of BeiDou B1C signals in the entire loop. This paper provides a detailed analysis of the performance of this algorithm from the perspectives of signal tracking accuracy, tracking sensitivity, and tracking time. Finally, the algorithm is implemented on an SDR platform, and practical tests are conducted in typical urban challenging scenarios to validate its effectiveness.

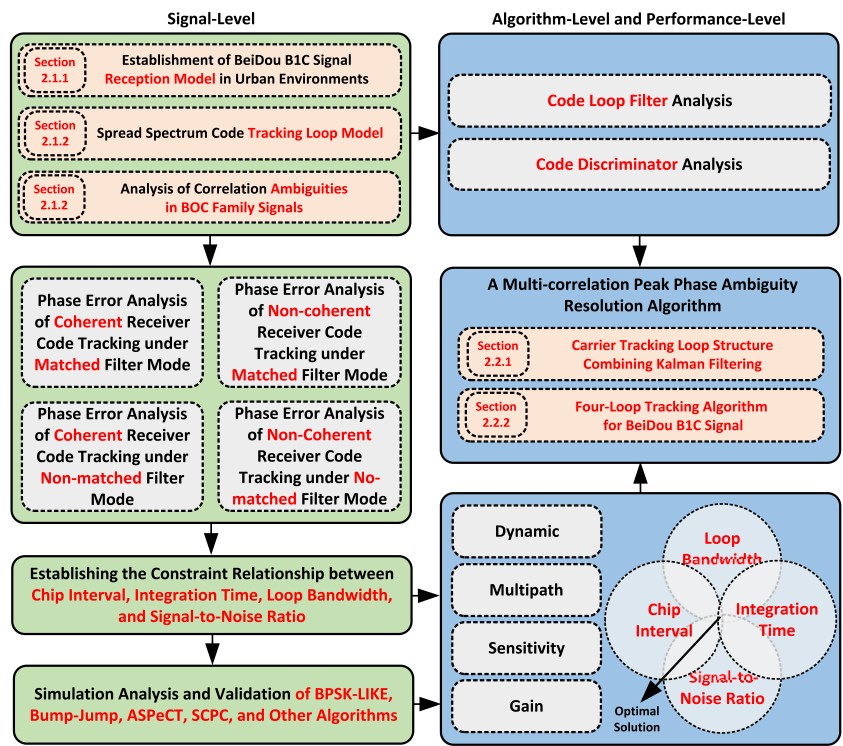

**Figure 1.** The Research Logic and Analysis Paradigm for the BeiDou B1C Signal and Synchronization Loop.

## 2.1. The Characteristics of BeiDou B1C Signal Structure

### 2.1.1. The Model and Structure of the BeiDou B1C Signal

In urban environments, satellite navigation signals play two crucial roles in the positioning process: distance measurement and data transmission. However, in such environments, there is often a trade-off between the ranging performance of a signal and its ability to transmit data accurately. For instance, in multipath environments, in order to achieve better ranging performance, a signal should ideally include more high-frequency components and concentrate its main energy at the band edges. On the other hand, to achieve a low bit error rate in data transmission, frequency components should be concentrated as much as possible at the center of the frequency band. Additionally, the ranging performance of the signal is influenced by whether the spread spectrum signal modulates the data and the associated data rate.

In the B1C signal of the third-generation BeiDou satellite navigation system, a design approach is employed that separates the data channel from the pilot channel. Different spreading sequences are used on these two channels, with navigation messages modulated on the data channel while the pilot channel remains unmodulated. The two channels are combined through multiplexing. According to the signal interface file [27] available on the official BeiDou website, the carrier frequency for the BeiDou B1C signal is 1575.42 MHz, with a bandwidth of 32.736 MHz and two orthogonal components. The signal structure and modulation schemes for each component are shown in Table 1.

**Table 1.** The B1C Signal Structure.

| Component | Modulation | | Phase Relationship | Power Ratio |
|---|---|---|---|---|
| $S_{B1C\_data}(t)$ | | SineBOC(1,1) | 0 | 1/4 |
| $S_{B1C\_pilot\_a}(t)$ | QMBOC(6,1,4/33) | SineBOC(1,1) | 90 | 29/44 |
| $S_{B1C\_pilot\_b}(t)$ | | SineBOC(6,1) | 0 | 1/11 |

The BeiDou B1C signal comprises two distinct components: the data component and the pilot frequency component. The data component is modulated using BOC, whereas the pilot frequency component is modulated using QMBOC.

The complex envelope of the BeiDou B1C signal is created by combining the data component, denoted as $S_{B1C\_data}$, and the pilot frequency component, denoted as $S_{B1C\_pilot}$, with a power ratio of 1:3. Following principles of IQ modulation in communication, the expression for the complex envelope of the B1C signal is as follows:

$$
\begin{aligned}
S_{B1C}(t) &= S_{B1C\_data}(t) + jS_{B1C\_pilot}(t) \\
S_{B1C\_data}(t) &= \tfrac{1}{2}D_{B1C\_data}(t) \cdot C_{B1C\_data}(t) \cdot SC_{B1C\_data}(t) \\
S_{B1C\_pilot}(t) &= \tfrac{\sqrt{3}}{2}C_{B1C\_pilot}(t) \cdot SC_{B1C\_pilot}(t)
\end{aligned}
\tag{1}
$$

where $D_{B1C\_data}$ represents the message data of the data component, $C_{B1C\_data}$ corresponds to the PRN code assigned to the data component, and $C_{B1C\_pilot}$ denotes the PRN code assigned to the pilot frequency component.

The data component of the B1C signal undergoes modulation using BOC, which is a modulation technique derived from the original BPSK-R with a square wave subcarrier. The subcarrier of the data component is denoted as $SC_{B1C\_data}$, and its mathematical expression is given by:

$$
SC_{B1C\_data}(t) = sign(\sin(2\pi f_{sc\_B1C\_a}t))
\tag{2}
$$

where $f_{sc\_B1C\_a}$ is 1.023 MHz.

The B1C pilot frequency component is $QMBOC(6,1,4/33)$, modulated by BOC(1,1) and BOC(6,1) distributed in a power ratio of 29:4 and modulated in two mutually orthogonal phases, respectively. Thus, the subcarrier expression of the pilot component $SC_{B1C\_pilot}$ is:

$$
SC_{B1C\_pilot}(t) = \sqrt{\frac{29}{33}}sign(\sin(2\pi f_{sc\_B1C\_a}t)) - j\sqrt{\frac{4}{33}}sign(\sin(2\pi f_{sc\_B1C\_b}t))
\tag{3}
$$

where $f_{sc\_B1C\_b}$ is 6.138 MHz.

In addition to the use of subcarriers, the pilot frequency component also employs a subcode with a period of 18 s and a code length of 1800 chips to modulate the ranging master code. Each subcode chip has a duration of 10 milliseconds, matching both the period of the ranging master code and the duration of a single bit in the navigation message within the data component. The complete expression for the B1C signal can be derived by organizing the above information:

$$
\begin{aligned}
s_{B1C}(t) &= \underbrace{s_{data}(t) + s_{pilot\_BOC(1,1)}(t)}_{BOC(1,1)} + \underbrace{s_{pilot\_BOC(6,1)}(t)}_{BOC(6,1)} \\
&= \tfrac{1}{2}D_{data}(t) \cdot C_{data}(t) \cdot \text{sign}\big(\sin(2\pi f_{C\_B1C\_a}t)\big) \\
&\quad + j\sqrt{\tfrac{29}{44}}C_{subpilot}(t) \cdot C_{pilot}(t) \cdot \text{sign}\big(\sin(2\pi f_{C\_B1C\_a}t)\big) \\
&\quad + \sqrt{\tfrac{1}{11}}C_{subpilot}(t) \cdot C_{pilot}(t) \cdot \text{sign}\big(\sin(2\pi f_{C\_B1C\_b}t)\big)
\end{aligned}
\tag{4}
$$

where $S_{pilot\_BOC(6,1)}(t)$ and $S_{pilot\_BOC(1,1)}(t)$ respectively represent the BOC(6,1) the BOC(1,1) component in the pilot frequency component. The entire B1C signal can be divided into two parts: the narrowband component BOC(1,1) and the wideband component BOC(6,1). These two components require different bandwidths during signal reception. The navigation message exists only in the narrowband component BOC(1,1), allowing for different reception strategies to be employed in the B1C signal reception scheme. QMBOC, employing the modulation of BOC(n,n) and BOC(m,n) components on two orthogonal phases, is a novel modulation approach for navigation signals that differs from both CBOC and Time-Multiplexed Binary Offset Carrier (TMBOC). In the design of BeiDou signals, this modulation scheme is adopted to ensure robust compatibility and coexistence with GPS

L1 and Galileo E1 signals at the same frequency, while also mitigating patent disputes. As shown in Figure 2, the spectrum of the BeiDou B1C signal is presented. In Figure 2, from left to right, we have depicted the BeiDou system's B2 signal, B3 signal, and B1 signal. The B2I signal from the BDS-2 is gradually being replaced by the B2a signal from the BDS-3. Additionally, the BDS-3 has introduced the B1C signal.

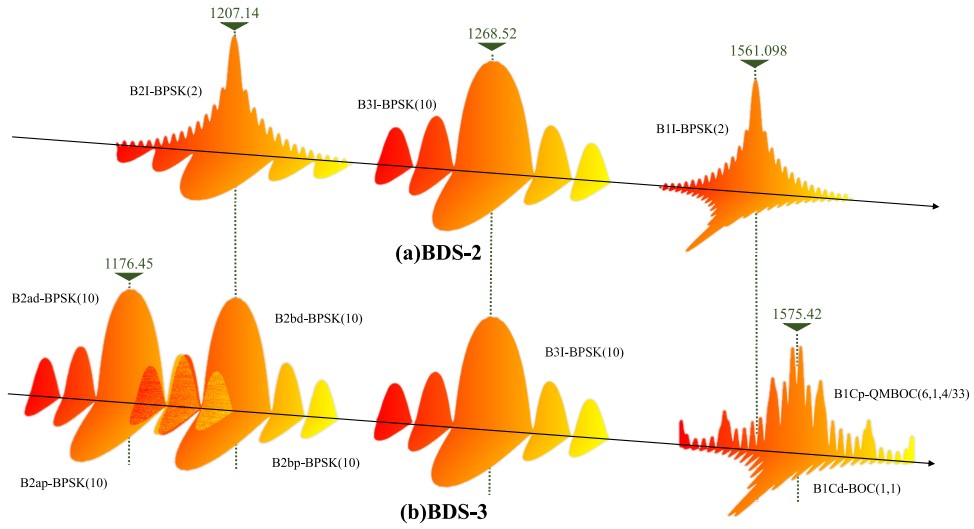

**Figure 2.** (**a**) Diagram of BeiDou-2 Signal Distribution. (**b**) Diagram of BeiDou-3 Signal Distribution.

### 2.1.2. Characteristics Analysis of the BeiDou B1C Signal

Based on the mathematical model and signal structure analysis presented in the previous section, it can be concluded that the data component of the BeiDou B1C signal adopts BOC(1,1) modulation. The pilot component can be considered as a mixture of BOC(1,1) and BOC(6,1) modulation signals. Therefore, the autocorrelation and frequency domain characteristics of the BeiDou B1C signal are directly related to the characteristics of BOC modulation signals. Compared to Binary Phase Shift Keying (BPSK) signals, modern satellite navigation BOC signals have longer PRN sequence periods and employ code families with better correlation properties. For example, the PRN code sequence period for BeiDou B1C signals is set to 10230, and it utilizes Weil codes with improved correlation properties [28]. The BOC signal $P_{BOC}(t)$ can be expressed as the product of a BPSK signal $P_{BPSK}(t)$ and a square wave signal:

$$P_{BOC}(t) = P_{BPSK}(t)\,\text{sign}(\sin(2\pi f_s t + \theta)) \tag{5}$$

where $f_s$ is the subcarrier frequency, and $\theta$ is the phase of the subcarrier. The commonly used values for $\theta$ are 0 and $\pi/2$, corresponding to sine phase and cosine phase, respectively.

The autocorrelation function of a Binary Offset Carrier (BOC) signal can be expressed as a combination of triangular functions and their time shifts. For a sine BOC signal, its autocorrelation function can be represented as follows:

$$R_{BOC_s}(\tau) = \Lambda_{T_s}(\tau) + \sum_{k=1}^{M-1}(-1)^k\left(1 - \frac{k}{M}\right)\Lambda_{T_s}(|\tau| - kT_s)$$
$$\Lambda_L(t) = \begin{cases} -\dfrac{|t|}{L}, & |t| \le L \\ 0, & \text{others} \end{cases} \tag{6}$$

For a cosine BOC signal, its autocorrelation function can be represented as follows:

$$R_{BOC_c}(\tau) = \Lambda_{T_s/2}(\tau) + \sum_{k=1}^{M-1}(-1)^k\left(1 - \frac{k}{M}\right)\Lambda_{T_s/2}(|\tau| - kT_s) +$$
$$\frac{1}{2M}\sum_{k=1}^{M-1}(-1)^k\Lambda_{T_s/2}(|\tau| - (2k-1)T_s/2) \tag{7}$$

where $M$ is the modulation order of the BOC(m,n) signal. $T$ is the interval between each correlation peak. $L$ represents the length of the time interval between the peaks of the triangular functions. According to Equations (5) and (6), the corresponding autocorrelation function curves can be obtained, as shown in Figure 3.

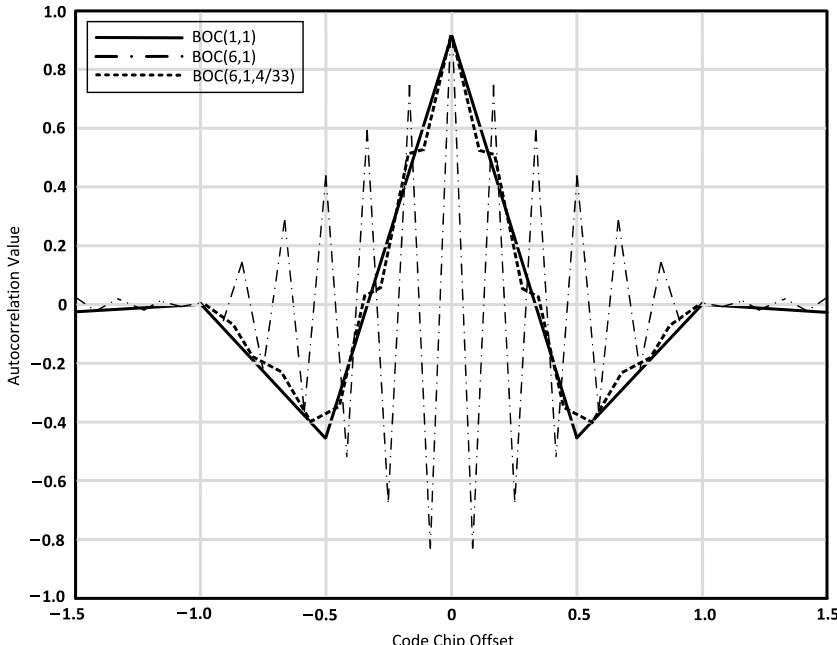

**Figure 3.** Comparison of Autocorrelation Functions for Different Modulation Modes.

As shown in Figure 3, it can be observed that the autocorrelation peak of the BOC(1,1) signal is narrower than that of the BPSK signal. However, there is a sidelobe present on both sides of the main peak, symmetrically positioned, with an amplitude equal to half of the main peak. The BOC(6,1) signal exhibits the sharpest peak, achieving the highest synchronization accuracy. However, it also has five sidelobes on both sides, with gradually decreasing amplitudes. As a result, it experiences significant ambiguity in the capture and tracking processes. The QMBOC(6,1,4/33) signal, due to its smaller weighting coefficients for the BOC(6,1) component, approximates the overall curve to the BOC(1,1) component that occupies the majority of the energy. The autocorrelation peak of the BOC signal is sharper than that of the BPSK-modulated signal, resulting in improved ranging performance and multipath suppression capability. However, it is susceptible to the influence of sidelobes, which can cause signal mislock at the peak positions of the sidelobes during the capture phase of the receiver, leading to false acquisition. In the tracking phase, the ambiguity of the tracking results due to the sidelobes can increase ranging errors. Therefore, it is necessary to mitigate the adverse effects of the sidelobes.

The content of various frequency components in a signal can be characterized by its power spectral density. The power spectral density of a spread spectrum signal determines the bandwidth required for its transmission and reception. Moreover, the power spectral density function can be used to analyze the ranging and demodulation performance of the signal in the presence of thermal noise and interference. Therefore, power spectral density is an important aspect in the design and performance analysis of spread spectrum modulation. For BOC modulation signals, Power Spectral Density (PSD) is also an important signal characteristic that reflects the design concept of this type of signal. Assuming an infinite Pseudorandom Noise (PRN) code period and ideal correlation properties with infinite

signal transmission and reception bandwidth, the normalized expression of the power spectral density for a BOC signal with a sinusoidal phase can be written as follows:

$$G_{BOC_s(f_s,f_c)}(f) = \begin{cases} \dfrac{\sin^2(\pi f T_c)\sin^2(\pi f T_s)}{T_c[\pi f \cos(\pi f T_s)]^2}, & M \text{ is an even number} \\ \dfrac{\cos^2(\pi f T_c)\sin^2(\pi f T_s)}{T_c[\pi f \cos(\pi f T_s)]^2}, & M \text{ is an odd number} \end{cases} \tag{8}$$

where $f_c$ and $f_s$ represent the code frequency and subcarrier frequency, respectively.

Considering the BOC modulation method used for the data and pilot components of the B1C signal, the PSD curves of the Sinusoidal BOC(1,1) and Sinusoidal BOC(6,1) signals are plotted within the frequency range of ±10 MHz around the center frequency. The results are shown in Figure 4.

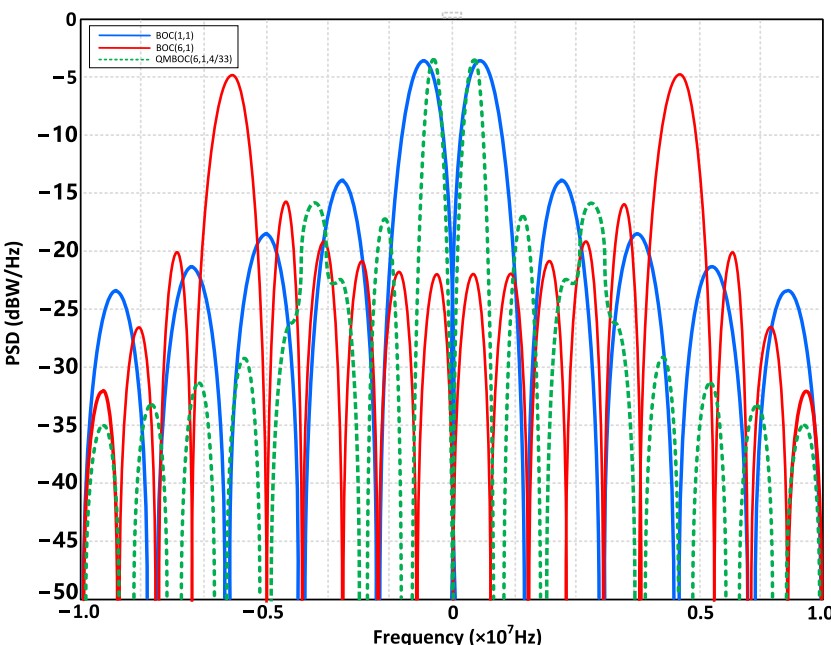

**Figure 4.** Power Spectral Density (PSD) Function of BOC Signal.

From Figure 4, it can be observed that the maximum value of the PSD of the BOC signal is no longer located at the carrier center frequency but is shifted to both sides. The main reason for this phenomenon is the effect of the subcarrier: a square wave subcarrier with a frequency of $f_s$ contributes strong harmonic components at $\pm f_s$. Considering the BOC signal as the product of a BPSK signal and a square wave subcarrier, in the frequency domain, the spectrum of the BOC signal can be seen as the secondary modulation of the BPSK signal, shifting the energy of the signal's main lobe to both sides at $\pm f_s$. Compared to the traditional BPSK signal with power concentrated at the center frequency, when the BOC signal and the BPSK signal share the same frequency, it does not cause significant interference to the reception of the traditional signal. Moreover, the design of BOC modulation is more flexible, providing a larger operational space. This effectively solves the problem of limited bandwidth resources caused by the congestion of signals in the L-band.

## 2.2. Multi-Correlation Peak Phase Deblurring Algorithm Design

### 2.2.1. Carrier Loop Design

Regarding the reception of the BeiDou B1C signal in urban environments, numerous challenges such as dynamics, obstructions, multipath, and interference are encountered. These scenarios introduce significant distortion and frequency offset in the received carrier signal, leading to substantial difficulties in signal tracking for the receiver. Currently,

carrier tracking algorithms can be mainly categorized into two types. The first type is the traditional carrier tracking algorithm, which utilizes a combination strategy between the phase-locked loop (PLL) and the frequency-locked loop (FLL) to achieve signal tracking. These algorithms primarily focus on two aspects: loop bandwidth and loop order. Dynamic loop bandwidth adjustment is commonly based on the consideration of system signal-to-noise ratio and correlation accumulation time, aiming to reduce the noise error in the loop by changing the bandwidth to thereby achieve carrier tracking [29–33]. On the other hand, variations in the loop order do not alter the noise characteristics of the loop, but instead affect the steady-state tracking error. The combination of FLL and PLL in carrier tracking algorithms presents challenges in determining the parameters. Only with appropriate values for the loop bandwidth can smaller tracking errors be achieved.

Another widely used type is the parameter estimation-based carrier tracking algorithm. Among them, the Kalman filter is an optimal estimation method suitable for both stationary and non-stationary stochastic processes [34–38]. It treats the signal process as a linear system driven by white noise. The relationship between the output and input is described by state equations. The estimation process utilizes the system's state equations, observation equations, and white noise excitation. If we assume that the filter system model and noise statistical characteristics are known and remain unchanged throughout the entire recursive processing, this algorithm can only be applicable when the model parameters strictly match the actual situation. Even slight changes in the model parameters can significantly degrade the tracking accuracy and may result in loss of lock in severe cases.

In response to the numerous challenges posed by the urban environment on carrier loop tracking, we propose a carrier phase tracking structure that combines Kalman filtering. This structure integrates the excellent signal tracking capabilities of FLL with the precise carrier phase prediction of Kalman filtering. By accurately estimating and dynamically updating the state and observation noises in the Kalman model, the proposed structure avoids model divergence and achieves improved performance and robustness in both the dynamic behavior and tracking accuracy of the carrier loop. The carrier loop structure is illustrated in Figure 5.

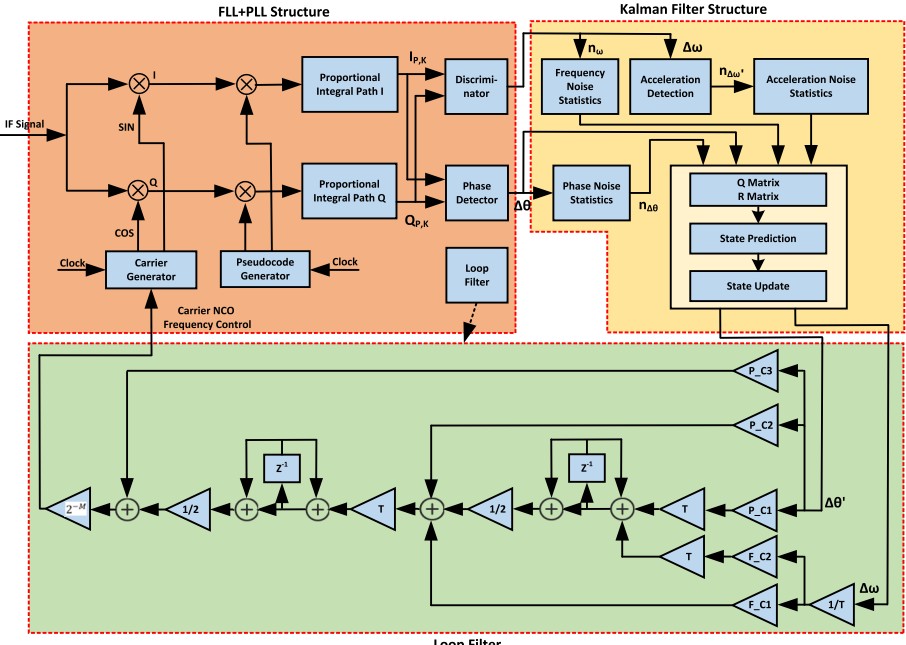

**Figure 5.** Carrier Loop Design Structure.

Upon the completion of receiver acquisition, the I and Q signals entering the carrier tracking loop undergo correlation accumulation. Subsequently, the two signals are as follows:

$$I_{P,k} = A_k D_k R(\Delta \tau_k) \frac{\sin(\pi \Delta f_k T_{coh})}{\pi \Delta f_k T_{coh}} \cos \theta_{e,k} + v_{I,k}$$
$$Q_{P,k} = A_k D_k R(\Delta \tau_k) \frac{\sin(\pi \Delta f_k T_{coh})}{\pi \Delta f_k T_{coh}} \sin \theta_{e,k} + v_{Q,k} \tag{9}$$

where $A_k$ represents the signal amplitude, $D_k$ represents the modulation data, $R(\Delta \tau_k)$ represents the autocorrelation function of pseudorandom codes, $\Delta f_k$ represents Doppler frequency shift estimation error, $T_{coh}$ represents pre-detection integration time, $\theta_{e,k}$ represents the phase difference, and $v_{I,k}$ and $v_{Q,k}$ represent Gaussian white noise.

The structure of the Kalman filter shown in Figure 5 is essentially a linear filter. Phase, frequency, and acceleration are filtered and preprocessed. Phase detector and frequency detector are still applicable to this structure, and they provide higher tracking accuracy within the phase or frequency range compared to FLL or PLL of the same order. The state equation of the Kalman filter in this case is expressed as:

$$\begin{bmatrix} \Delta \theta(k+1) \\ \Delta \omega_0(k+1) \\ \Delta \omega_1(k+1) \end{bmatrix} = \begin{bmatrix} 1 & T & T^2/2 \\ 0 & 1 & T \\ 0 & 0 & 1 \end{bmatrix} \begin{bmatrix} \Delta \theta(k) \\ \Delta \omega_0(k) \\ \Delta \omega_1(k) \end{bmatrix} + \begin{bmatrix} f_{IF} T \\ 0 \\ 0 \end{bmatrix} + W \tag{10}$$

where $T$ represents the loop update time. $W$ represents the state noise, which can be expressed in the form of white noise:

$$E[W_k] = 0$$
$$E[W_k W_k^T] = Q_k \tag{11}$$

where the state noise matrix $Q_k$ is related to the dynamics of the carrier. The observation equation is expressed as:

$$\Delta \theta(k) = \begin{bmatrix} 1 & 0 & 0 \end{bmatrix} \begin{bmatrix} \Delta \theta(k) \\ \Delta \omega_0(k) \\ \Delta \omega_1(k) \end{bmatrix} + V_k \tag{12}$$

The observation noise is represented in the form of white noise:

$$E[V_k] = 0$$
$$E[V_k V_k^T] = R_k \tag{13}$$

The computation process of the Kalman filter consists of two major steps: prediction and update.

The prediction step includes:

$$P_{k|k-1} = \Phi_{k,k-1} P_{k-1} \Phi_{k,k-1}^T + \Gamma_{k-1} C_{w_{k-1}} \Gamma_{k-1}^T$$
$$\hat{s}_{k|k-1} = \Phi_{k,k-1} \hat{s}_{k-1} \tag{14}$$

$\Gamma_{k-1}$ is referred to as the system control matrix at time $k$, which reflects the extent to which the disturbance noise vector $w_{k-1}$ influences the system state vector. $\Phi_{k,k-1}$ is referred to as the one-step state transition matrix. $P_{k|k-1}$ is the one-step predicted mean square error matrix.

The update step includes:

$$K_k = P_{k|k-1} H_k^T \left( H_k P_{k|k-1} H_k^T + C_{n_k} \right)^{-1}$$
$$P_k = (I - K_k H_k) P_{k|k-1}$$
$$\hat{s}_k = \hat{s}_{k|k-1} + K_k \left( x_k - H_k \hat{s}_{k|k-1} \right) \tag{15}$$

$H_k$ is the observation matrix at time $k$, and $n_k$ is the observation noise vector at time $k$. The steady-state bandwidth of the loop is only determined by $Q_k$ and $R_k$, which are set based on the noise characteristics and dynamic properties of the satellite signal.

In the face of various challenges posed by urban environments to carrier loop tracking, our proposed carrier loop structure can reduce the bandwidth of PLL and FLL, suppress signal noise energy, and improve multipath resistance when dealing with blockage and multipath environments. In the case of highly dynamic scenarios, the loop bandwidth can be increased, and the thermal noise introduced by the larger loop bandwidth can be addressed by the jointly used Kalman filtering method. The Kalman filter, as an optimal filter, can ensure loop stability in this scenario based on the establishment of an appropriate dynamic GNSS signal carrier tracking model.

### 2.2.2. Code Loop Design

The tracking and synchronization of the BeiDou B1C signal are achieved through the mutual assistance of the carrier loop and code loop. However, the multi-peak autocorrelation characteristic of the BeiDou B1C signal introduces ambiguity issues in the code loop. Furthermore, the B1C signal is comprised of two components: the data component and the pilot component. This division results in the utilization of only a portion of the total signal energy for tracking a single component. Consequently, the decrease in target signal energy inevitably leads to a decline in tracking accuracy. It is evident that tracking either the data component or the pilot component alone is not the optimal tracking strategy.

In the process of tracking B1C signals, phase ambiguity may manifest due to the existence of multiple zero-crossing points in the phase detection curve, as illustrated in Figure 6. To address this problem, an Early Minus Late Amplitude (EMLA) phase detection method was employed to analyze the phase of the BOC(1,1), BOC(6,1), and QMBOC(6,1,4/33) signals. Based on the phase detection results shown in Figure 6, the following conclusion can be drawn:

1. The higher the modulation order of the BOC signal, the higher the level of ambiguity (the greater the number of zero-crossings in phase detection).
2. As the correlation interval becomes smaller, the phase detection accuracy increases.

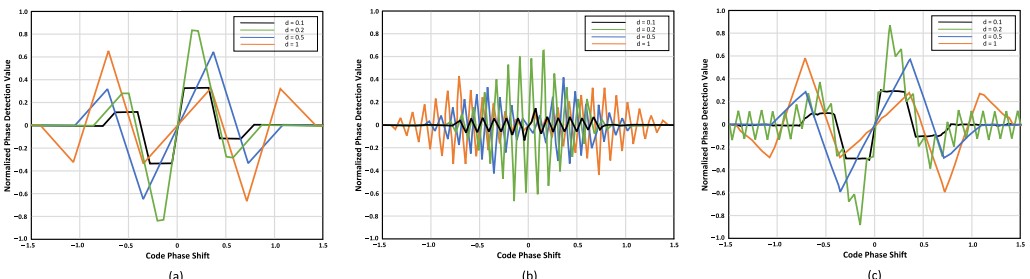

**Figure 6.** (**a**) Phase Detection Result of BOC(1,1). (**b**) Phase Detection Result of BOC(6,1). (**c**) Phase Detection Result of QMBOC(6,1,4/33).

To address the tracking ambiguity of the B1C signals and improve tracking accuracy in urban scenarios, and to make full use of different component information, we propose a pseudo-code tracking loop structure for B1C signals. This structure utilizes the accuracy errors in phase estimation from the pseudo-code tracking loop, carrier tracking loop, and sub-carrier tracking loop to compute the correct phase point, resulting in a stable tracking loop. Figure 7 depicts the loop structure that we have proposed.

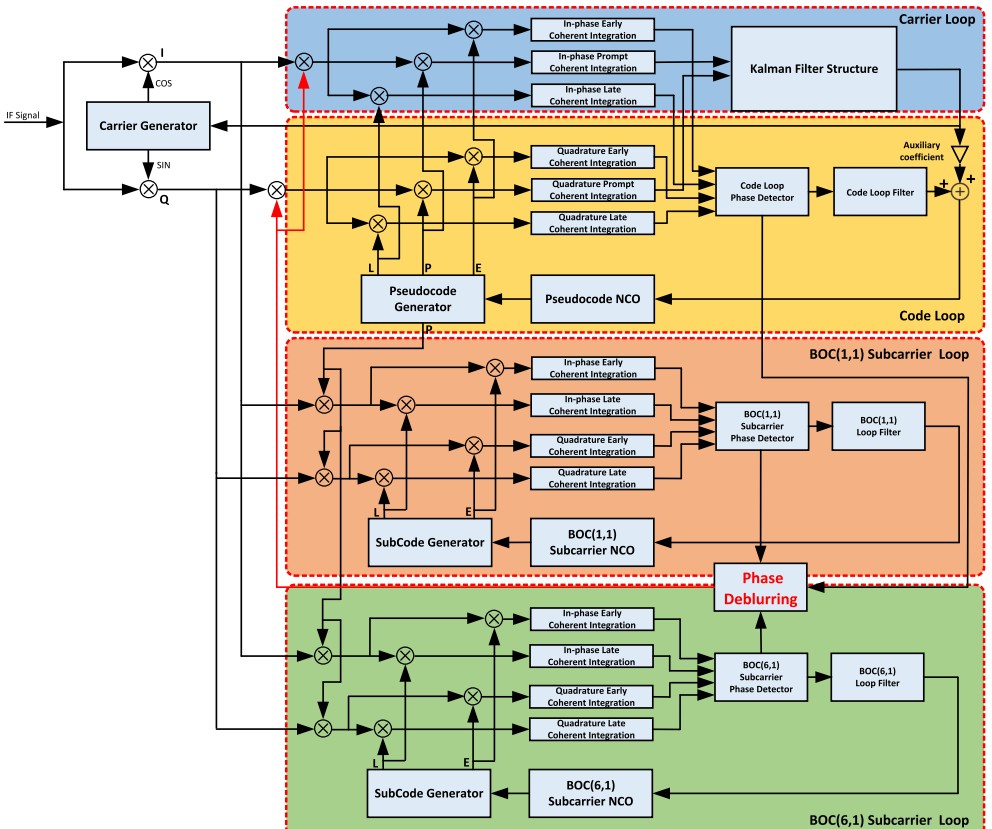

**Figure 7.** Multi-correlation Peak Phase Deblurring Algorithm Structure.

In Figure 7, in addition to the loop carrier tracking loop and the code tracking loop, there are subcarrier tracking loops for the BOC(1,1) and BOC(6,1) signals. The intermediate frequency signal input is multiplied by the carrier NCO to remove the carrier, and then divided into two branches. One branch is multiplied by the prompt branch subcarrier and enters the code tracking loop. The code loop has three branches: early, prompt, and late. Among them, the prompt branch output serves as the carrier phase detection result and can be represented as:

$$u_{PP} = \int_T s_{IF}(t) e^{-j(2\pi \tilde{f} + \tilde{\theta})} \times c(t - \tilde{\tau}_c) \times sc(t - \tilde{\tau}_s) \mathrm{d}t \tag{16}$$

where $s_{IF}(t)$ is intermediate frequency signal, $\tilde{f}$ is local carrier frequency, $\tilde{\theta}$ is local carrier phase, and $\hat{\tau}$ is code delay. The integration clearing operation is performed on the early-late branch of the code loop, and the result can be represented as:

$$\begin{aligned} u_{EP} &= \int_T s_{IF}(t) e^{-j(2\tilde{f} + \tilde{\theta})} \times c(t - \tilde{\tau}_c + d_c) \times sc(t - \tilde{\tau}_s) \mathrm{d}t \\ u_{LP} &= \int_T s_{IF}(t) e^{-j(2\tilde{f} + \tilde{\theta})} \times c(t - \tilde{\tau}_c - d_c) \times sc(t - \tilde{\tau}_s) \mathrm{d}t \end{aligned} \tag{17}$$

where $d_c$ is code correlation spacing. The integration principle in the subcarrier loop is similar to that of the code loop. The result of the integration can be represented as:

$$\begin{aligned} u_{PE} &= \int_T s_{IF}(t) e^{-j(2\pi \tilde{f} + \tilde{\theta})} \times c(t - \tilde{\tau}_c) \times sc(t - \tilde{\tau}_s + d_s) \mathrm{d}t \\ u_{PL} &= \int_T s_{IF}(t) e^{-j(2\pi \tilde{f} + \tilde{\theta})} \times c(t - \tilde{\tau}_c) \times sc(t - \tilde{\tau}_s - d_s) \mathrm{d}t \\ u_{PE_{61}} &= \int_T s_{IF}(t) e^{-j(2\pi \tilde{f} + \tilde{\theta})} \times c(t - \tilde{\tau}_c) \times sc(t - \tilde{\tau}_s + d_{s_{61}}) \mathrm{d}t \\ u_{PL_{61}} &= \int_T s_{IF}(t) e^{-j(2\tilde{f} + \tilde{\theta})} \times c(t - \tilde{\tau}_c) \times sc(t - \tilde{\tau}_s - d_{s_{61}}) \mathrm{d}t \end{aligned} \tag{18}$$

where $d_s$ is the spacing of the BOC(1,1) subcarrier correlator, and $d_{s61}$ is the spacing of the BOC(6,1) subcarrier correlator.

The correlation between the code tracking loop, carrier tracking loop, and subcarrier tracking loop can be described using the following function. The signal correlation process is represented as:

$$
\begin{aligned}
u &= \frac{1}{T} \int_0^T r_{\text{BOC}}(t) \times \cos(\omega_0 t + \hat{\varphi}) \times s(t - \hat{\tau}^*) \times c(t - \hat{\tau}) \mathrm{d}t \times d \\
&\approx \cos(\varphi - \hat{\varphi}) \times \text{trc}(\hat{\tau}^* - \tau) \times \Lambda(\hat{\tau} - \tau) \times d
\end{aligned}
\tag{19}
$$

where $T$ is integration time, $r_{\text{BOC}}(t)$ is the input signal that includes phase delay and carrier phase, $s(t - \hat{\tau}^*)$ is the local subcarrier with delay estimation, $c(t - \hat{\tau})$ is the local pseudocode with delay estimation, $\hat{\tau}^*$ and $\hat{\tau}$ are delay estimation. The correlation function $\text{trc}(\cdot)$ for the subcarrier is constructed by a continuous triangular cosine function. $\Lambda(\cdot)$ represents the correlation function for the pseudocode; $d$ represents the navigation message; and $\hat{\varphi}$ represents the phase offset. If we ignore the navigation message and assume complete carrier synchronization, the simplification of Equation (19) would be:

$$
q(\hat{\tau}, \hat{\tau}^*) \approx \text{trc}(\hat{\tau}^* - \tau) \times \Lambda(\hat{\tau} - \tau)
\tag{20}
$$

In the $q$ function, the independent variables represent the delays of the code and subcarrier phase, assuming the true delay offset $\tau$ is zero.

In Equation (20), when $\hat{\tau} = \tau$, the correlation function becomes $q(0, \hat{\tau}^*) = \text{trc}(\hat{\tau}^* - \tau)$. Indeed, this represents the correlation in the subcarrier domain. When $\hat{\tau}^* = \tau$, the correlation function becomes $q(\hat{\tau}, 0) = \Lambda(\hat{\tau} - \tau)$, representing the correlation in the pseudocode domain. The correlation function curve of the QMBOC signal is shown in Figure 8.

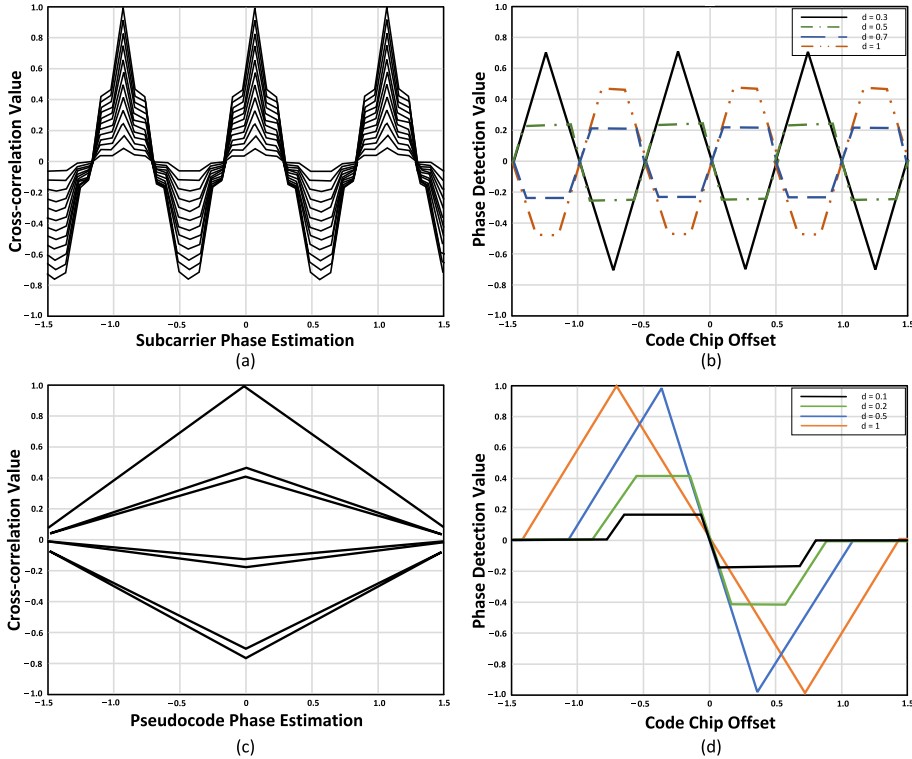

**Figure 8.** (**a**) Cross-correlation Values in the Subcarrier Domain. (**b**) Phase Detection Values in the Subcarrier Domain. (**c**) Cross-correlation Values in the Pseudocode Domain. (**d**) Phase Detection Values in the Pseudocode Domain.

In Figure 8b, the subcarrier adopts EMLA phase detection, and the maximum range of phase detection is obtained with $\delta = 0.3$. The size of the phase detection range can be dynamically adjusted to meet different requirements under different dynamics. When the loop is in a large dynamic range, adjusting the correlation interval can match the dynamics and achieve the maximum subcarrier tracking range. If high tracking accuracy is required, the correlation interval can be adjusted to obtain better tracking accuracy.

In Figure 8d, when $d \leq 0.5$, the slope of the pseudocode phase detection remains consistent. The larger the interval, the greater the amplitude of the phase detection curve. When $d = 0.5$, the maximum amplitude range of the phase detection is obtained. When $d = 1$, the amplitude remains constant, but the slope value decreases. This type of situation is suitable for tracking the impact of dynamics on the pseudocode, particularly under certain dynamic conditions.

QMBOC signal exhibits distinct characteristics in the pseudocode and subcarrier domains:

1. In the pseudocode domain, the correlation peaks only reflect the pseudocode correlation, with sharp peaks but low precision and no ambiguity. Due to the influence of subcarrier correlation values, there are pseudocode correlation values both above and below the peak.
2. On the other hand, in the subcarrier domain, the subcarrier has a higher frequency, resulting in narrower peaks and higher phase detection accuracy. However, there is ambiguity present, leading to multiple peaks.

In the above analysis, the pseudocode and subcarrier phase estimations were considered as two independent parameters. However, during the signal synchronization process, the subcarrier phase is synchronized with the pseudocode phase. Due to the existence of ambiguity, the subcarrier phase estimation is shifted by integer multiples of the subcarrier period, leading to the following equation:

$$\hat{\tau}_s = \hat{\tau}_c + NT_s \tag{21}$$

where $N$ is an integer, and $T_s$ represents the subcarrier period. The convergence estimation of the subcarrier phase loop is provided by $\hat{\tau}_c$. The final pseudorange measurement estimation is achieved through the calculation of mean squared error. Such a phase detector fully utilizes the narrow peak high precision capability of BOC modulation and exhibits smooth characteristics.

The comparison between the pseudocode and the subcarrier phase detection of the pilot component is shown in Figure 9.

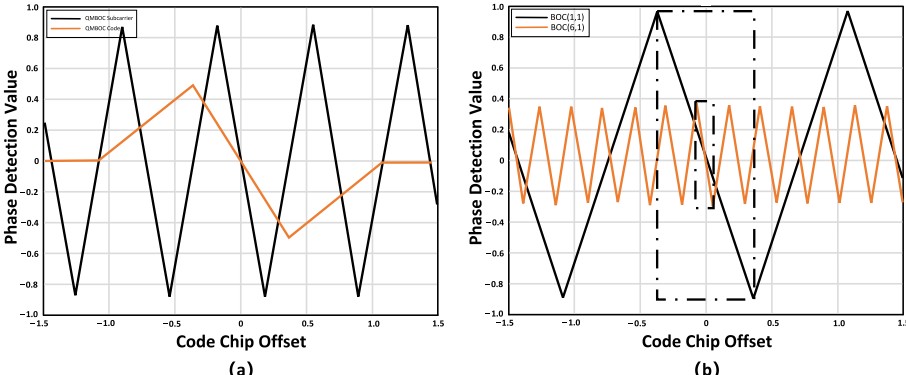

**Figure 9. (a)** Comparison of Pseudocode and Subcarrier Phase Detection. **(b)** Phase Detection for Different Subcarriers.

In Figure 9a, the intersection of the two curves of the subcarrier and pseudocode at the zero point represents the target position for tracking, while the other points are false lock points. Figure 9 effectively reflects the fundamental characteristics of pseudocode and subcarrier phase detection. It also indicates that the organic combination of the two can

achieve high-precision tracking of BOC family signals and eliminate the existing ambiguity issues. Based on the above analysis, the following conclusions can be drawn:

$$\hat{\tau}' = \hat{\tau}_s + \text{round}\left(\frac{\hat{\tau}_s - \hat{\tau}_c}{T_s}\right) \cdot T_s \tag{22}$$

By continuously comparing with the estimated rounded difference value $\hat{\tau}_c$, the final estimation of $\hat{\tau}'_s = \hat{\tau}_s$ is obtained, which eliminates the whole-cycle ambiguity in Subcarrier Lock Loop (SLL) and incorporates it into the phase estimation value $\hat{\tau}'$ synchronized to the pseudocode, achieving unambiguous high-precision tracking. The tracking accuracy is achieved through subcarrier phase detection. The subcarrier phase detection comparison of the QMBOC components is shown in Figure 9b. The dashed large box indicates the phase detection range for the BOC(1,1) subcarrier, and the dashed small box indicates the phase detection range for the BOC(6,1) subcarrier.

In the absence of baseband signals and noise, the Non-coherent Early-Late Power (NELP) discriminator and Non-coherent Dot Product discriminator are used to distinguish DLL and SLL. The formulas are as follows:

$$\begin{aligned} R_{\text{DLL}} &= q(\hat{\tau} + \Delta_c/2, \hat{\tau}^*)^2 - q(\hat{\tau} - \Delta_c/2, \hat{\tau}^*)^2 \\ R_{\text{SLL}} &= (q(\hat{\tau}, \hat{\tau}^* + \Delta_s/2) - q(\hat{\tau}, \hat{\tau}^* - \Delta_s/2)) \times q(\hat{\tau}, \hat{\tau}^*) \end{aligned} \tag{23}$$

where $\hat{\tau}$ represents the delay of the pseudocode, $\hat{\tau}^*$ represents the delay of the subcarrier, $\Delta_c$ represents the correlation interval of the pseudocode, and $\Delta_s$ represents the correlation interval of the subcarrier.

DLL and SLL in dual-loop tracking are mutually influenced, and there is a certain coupling between them. The translation of multiple zero-crossing points in the subcarrier dimension into the pseudocode dimension amplifies the estimation error in the pseudocode dimension. The error accumulation from coarse pseudocode phase estimation to fine estimation deteriorates the steady-state tracking error of SLL, and it may even prevent entering the steady-state tracking state. To eliminate the adverse effects of coupling, on the basis of Equation (23), the Early-Late discriminator is added as an additional dimension parameter, forming a dual discriminator:

$$\begin{aligned} R_{\text{DLL}} &= q(\hat{\tau} + \Delta_c/2, \hat{\tau}^* + \Delta_s/2)^2 + q(\hat{\tau} + \Delta_c/2, \hat{\tau}^* - \Delta_s/2)^2 \\ &\quad - q(\hat{\tau} - \Delta_c/2, \hat{\tau}^* + \Delta_s/2)^2 - q(\hat{\tau} - \Delta_c/2, \hat{\tau}^* - \Delta_s/2)^2 \\ R_{\text{SLL}} &= \left( \begin{array}{c} q(\hat{\tau} + \Delta_c/2, \hat{\tau}^* + \Delta_s/2) + q(\hat{\tau} + \Delta_c/2, \hat{\tau}^* - \Delta_s/2) \\ -q(\hat{\tau} - \Delta_c/2, \hat{\tau}^* + \Delta_s/2) - q(\hat{\tau} - \Delta_c/2, \hat{\tau}^* - \Delta_s/2) \end{array} \right) \times q(\hat{\tau}, \hat{\tau}^*) \end{aligned} \tag{24}$$

In quadruple-loop tracking, the estimation values are generated according to the following proportions:

$$\hat{\tau}'' = a \times \left( \hat{\tau}_{s_{11}} + \text{round}\left(\frac{\hat{\tau}_{s_{11}} - \hat{\tau}_c}{T_{s_{11}}}\right) \cdot T_{s_{11}} \right) + (1 - a) \cdot \left( \hat{\tau}_{s_{61}} + \text{round}\left(\frac{\hat{\tau}_{s_{61}} - \hat{\tau}_c}{T_{s_{61}}}\right) \cdot T_{s_{61}} \right) \tag{25}$$

where $\hat{\tau}_{s_{11}}$ and $\hat{\tau}_{s_{61}}$ are the phase estimation values for the BOC(1,1) signal and the BOC(6,1) signal, respectively. $T_{s_{11}}$ and $T_{s_{61}}$ are the subcarrier period values for the BOC(1,1) signal and the BOC(6,1) signal, respectively, and $a$ is the power coefficient. The steps of the code phase deblurring algorithm presented above are illustrated in Figure 10.

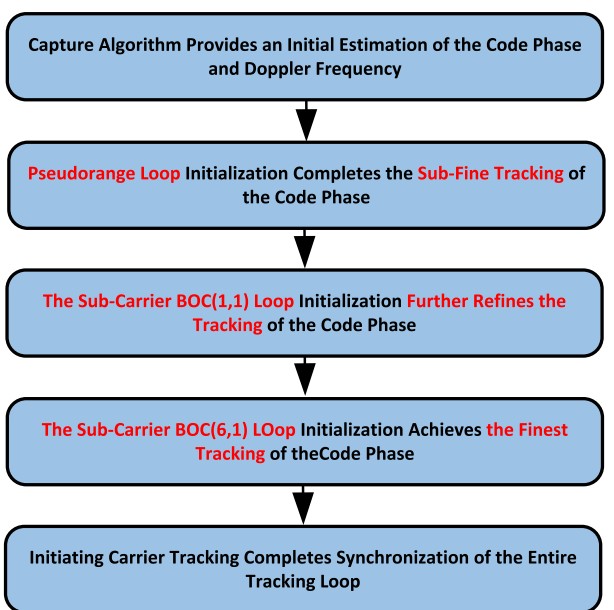

**Figure 10.** The Steps of the Code Phase Deblurring Algorithm.

## 3. Results

### 3.1. Simulation and Analysis

Upon comparing the tracking methods under the code correction condition in the QMBOC signals, the obtained results are depicted in Figure 11.

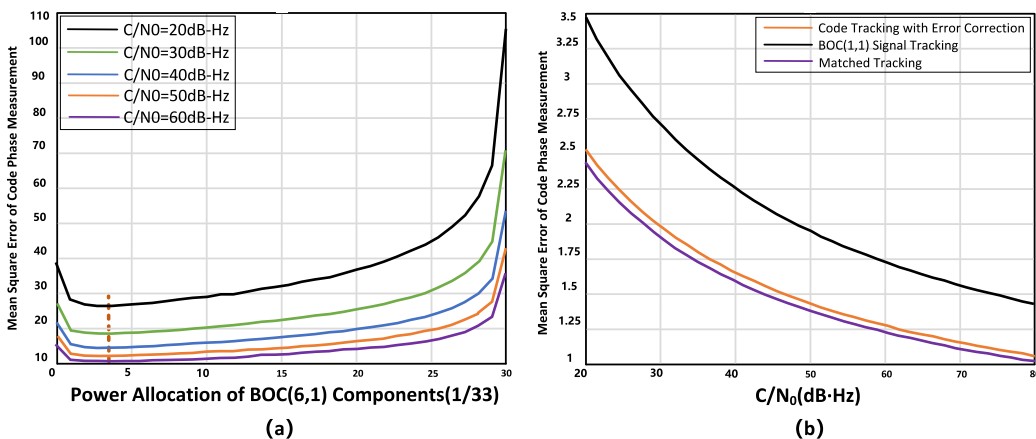

**Figure 11.** (**a**) Comparative Analysis of Code Correction Parameters in QMBOC Signals. (**b**) Comparative Analysis of Tracking Methods for QMBOC Signals.

According to the tracking error results obtained at different signal-to-noise ratios (CNR) shown in Figure 11a, it is evident that the BOC(6,1) component with a subcarrier ratio of $a = 4/33$ exhibits the lowest tracking error. Furthermore, the comparison presented in Figure 11b indicates that the tracking error in code correction tracking is greater than that in matched tracking and smaller than the tracking error in BOC(1,1) tracking. Hence, setting $a = 4/33$ yields corresponding quadruple-loop tracking results, effectively improving the accuracy of signal tracking. These findings suggest that selecting the BOC(6,1) component with a $a = 4/33$ configuration can optimize the tracking performance by minimizing the tracking error. This information is valuable for the design and implementation of tracking systems aiming to achieve higher accuracy in signal tracking.

To investigate the effects of these signal characteristics, this study designed two experimental groups: a noise-free group and a control group with the presence of noise. Under noise-free conditions, simulated B1C data with a duration of 5 s and PRN number

20 were generated. The initial code phase was set to 4120 chips, and the initial Doppler frequency was set to 800 Hz. Additionally, a linearly changing Doppler frequency shift of 4 Hz per second was introduced into the signal. To simulate urban scenarios, additional simulation data were generated with an SNR of −20 dB as a comparative test. Keeping other parameters constant, this simulated data serve as a contrast to evaluate the performance of our designed B1C signal tracking algorithm. Separate tests were conducted on the pilot and data components of the two data sets. The code phase discriminator spacing was set to 0.2 chips. Based on the tracking results from these tests, the following sections present a comparative analysis from four aspects: coherent integration values of the in-phase and quadrature branches, non-coherent integration amplitudes of the Early-Prompt-Late (EPL) branches, Doppler frequency, and code phase. The simulation test parameters are presented in Table 2.

**Table 2.** Simulation Test Parameters.

| Parameter | Value |
|---|---|
| Sampling Rate | 120 MHz |
| Signal Duration | 5 s |
| PRN Number | 20 |
| Initial Code Phase | 4120th chip |
| Initial Doppler Frequency | 800 Hz |
| Doppler Frequency Shift | 4 Hz per second |
| Signal-to-Noise Ratio (SNR) | −20 dB |
| Code Phase Correlation Interval | 0.5 chip |
| BOC (1,1) Subcarrier Correlation Interval | 0.5 chip |
| BOC (6,1) Subcarrier Correlation Interval | 0.2 chip |
| Loop Noise Bandwidth | 5 Hz |
| Carrier Loop Bandwidth | 20 Hz |
| Damping Coefficient | 0.707 |

Figure 12 depicts the distribution of the in-phase (I) and quadrature (Q) values, which exhibit two distinct clusters, a characteristic indicative of effective satellite signal tracking. Under noise-free conditions, the I and Q values of the B1C-simulated signal tracking results exhibit relatively concentrated distributions, with the I amplitudes of the pilot component approximately twice that of the data component. However, under the noise conditions with SNR = −20 dB, the amplitudes and clustering of the I and Q value distributions noticeably decrease. Furthermore, the data component with lower energy exhibits a more dispersed pattern.

Based on Figure 13, it is evident that under noise-free conditions, the tracking loop exhibits a transient fluctuation before quickly achieving stable tracking of the Doppler frequency variation. The estimated Doppler frequency demonstrates a linear trend, reaching a value of 16 Hz at 4 s, which is consistent with the parameter settings used during the simulation of the signal. This alignment validates the successful tracking capability of the loop for the signal. Under the noise conditions with an SNR of −20 dB, the tracking loop still follows the correct trend when tracking the Doppler frequency; however, fluctuations occur due to the influence of noise. Notably, the data component exhibits relatively larger tracking errors compared to the pilot component.

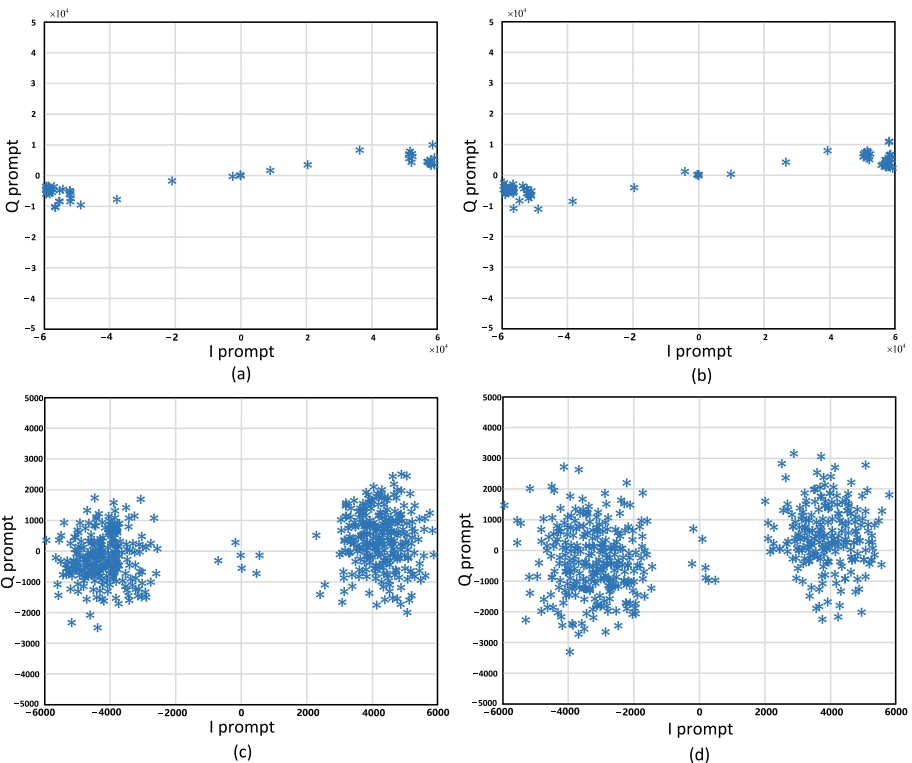

**Figure 12.** Comparative Analysis of In-Phase (I) and Quadrature (Q) Value Distributions for Separate Tracking of B1C Pilot and Data Components. (**a**) Noise-free pilot branch. (**b**) Noise-free data branch. (**c**) SNR = −20 dB pilot branch. (**d**) SNR = −20 dB data branch.

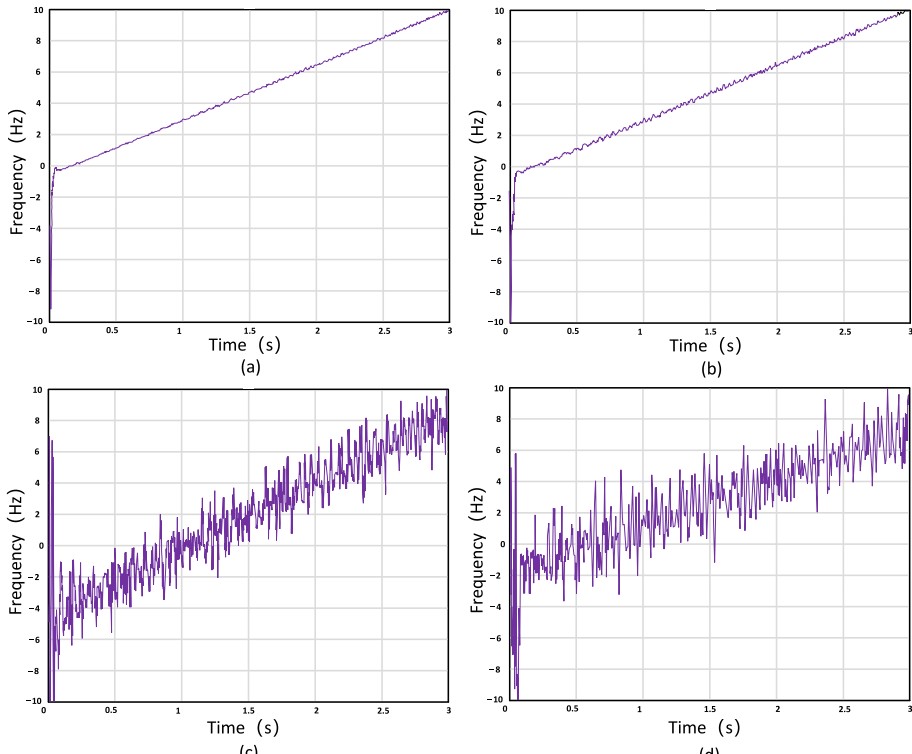

**Figure 13.** Comparative Analysis of Doppler Estimation for Separate Tracking of B1C Pilot and Data Components. (**a**) Noise-free pilot branch. (**b**) Noise-free data branch. (**c**) SNR = −20 dB pilot branch. (**d**) SNR = −20 dB data branch.

From Figure 14, it is evident that the multi-peak autocorrelation characteristics of the B1C signal significantly impact the tracking loop, resulting in increased code phase tracking errors. Under noise-free conditions, the tracking loop maintains stable tracking of the code phase after a transient fluctuation, keeping the estimation error near 0 chips. This stability allows the prompt path (P-path) to remain locked onto the main correlation peak. However, under the noise conditions with an SNR of −20 dB, the influence of noise on the correlation curve may cause the code phase tracking errors to exceed the ideal control range of the code phase discriminator. As a result, the impact of side peaks on the main peak becomes more pronounced. This is reflected in the significant fluctuations in the code phase tracking errors, particularly in the tracking of the data component. These fluctuations not only introduce larger tracking errors, but also greatly reduce the stability and reliability of the tracking loop.

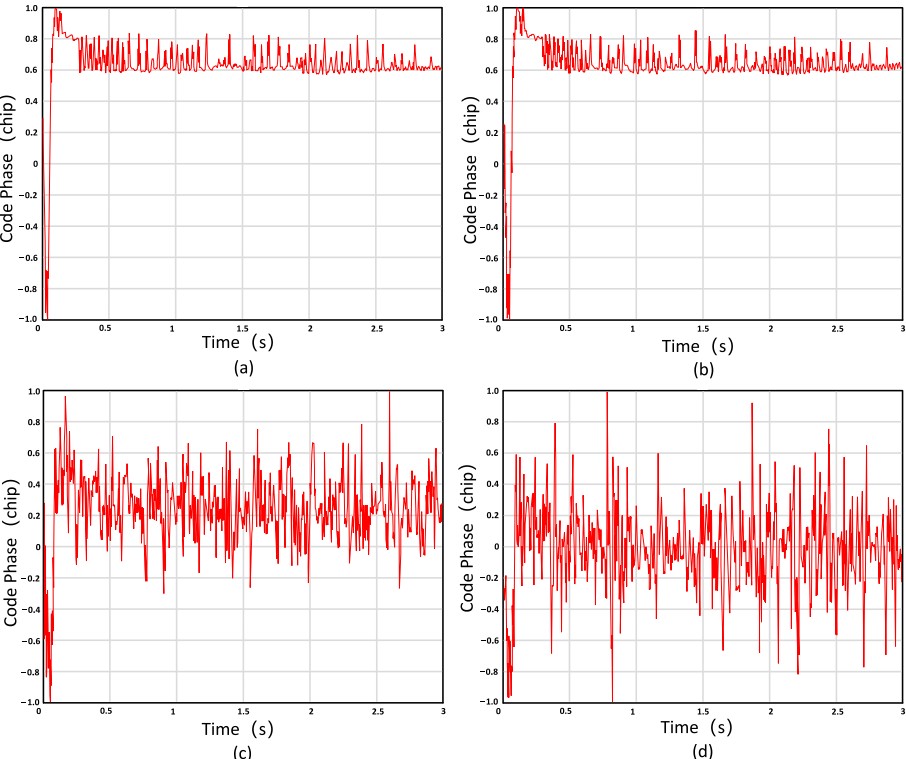

**Figure 14.** Comparative Analysis of Code Phase Estimation for Separate Tracking of B1C Pilot and Data Components. (**a**) Noise-free pilot branch. (**b**) Noise-free data branch. (**c**) SNR = −20 dB pilot branch. (**d**) SNR = −20 dB data branch.

The analysis of subcarrier tracking performance for the BOC(1,1) and BOC(6,1) components is further pursued. As depicted in Figure 9, there are significant differences in the subcarrier correlator spacing between the BOC(1,1) and the BOC(6,1) signals. The selection of an appropriate sampling frequency is crucial to meet the requirements of narrow correlation spacing while accommodating the wider correlation spacing demanded by high dynamic tracking environments, aiming to optimize the phase tracking range. The comparative analysis of subcarrier component errors for different chip intervals under an SNR of −20 dB is shown in Figure 15.

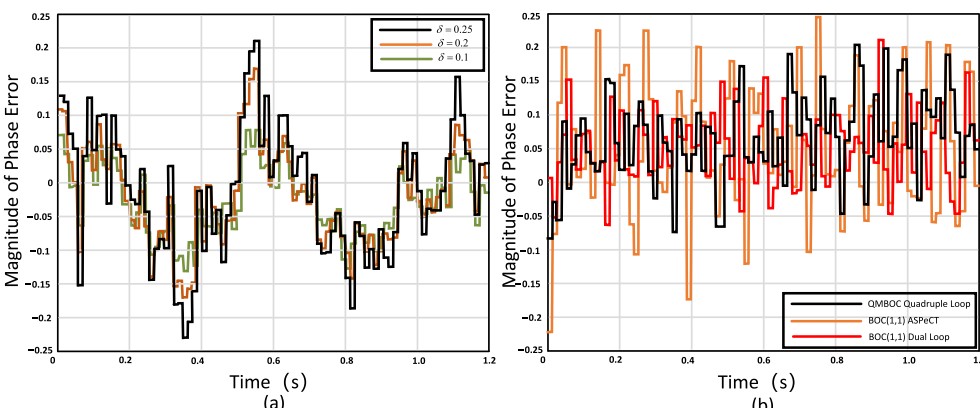

**Figure 15.** Comparison of Subcarrier Phase Error Under Different Chip Intervals. (**a**) Comparison of subcarrier intervals for BOC(1,1). (**b**) Comparison of subcarrier intervals for BOC(6,1).

From Figure 15, it can be observed that the subcarrier tracking errors for the BOC(1,1) signals exhibit similar trends across different correlation spacings. However, BOC(6,1) signals fail to meet the requirements for reliable signal tracking, as their tracking performance is unstable. Further improvements are needed, especially for applications with low signal-to-noise ratios. Additionally, considering that the BOC(6,1) component accounts for only 4/33 of the QMBOC signal, the dual-loop tracking approach commonly used for BOC(1,1) signals is sufficient to achieve synchronization among the components of the QMBOC signal.

Based on the aforementioned analysis, a subcarrier spacing of d = 0.5 chips is selected for BOC(1,1) signals, while a subcarrier spacing of d = 0.2 chips is chosen for BOC(6,1) signals. The pseudocode correlation spacing is set at d = 0.5 chips. A comparative analysis is performed with the non-matched tracking method for BOC(1,1) signals, and the resulting tracking phase errors for the carrier, pseudocode, and subcarrier are shown in Figure 16.

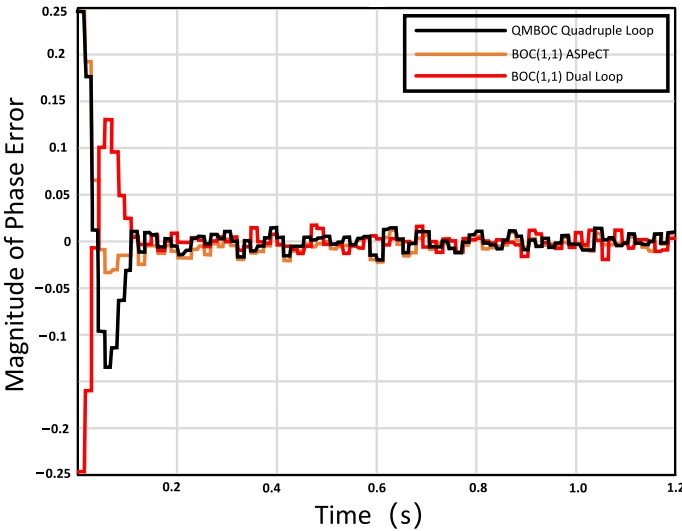

**Figure 16.** Carrier Loop Tracking Results.

Compared to the ASPeCT method for BOC(1,1), the dual-loop and quadruple-loop tracking methods for BOC(1,1) demonstrate significantly lower mean errors, approximately 40% smaller, when achieving stable tracking at t = 0.1 s. This indicates a higher tracking accuracy. The comparison of pseudocode tracking accuracy is shown in Figure 17.

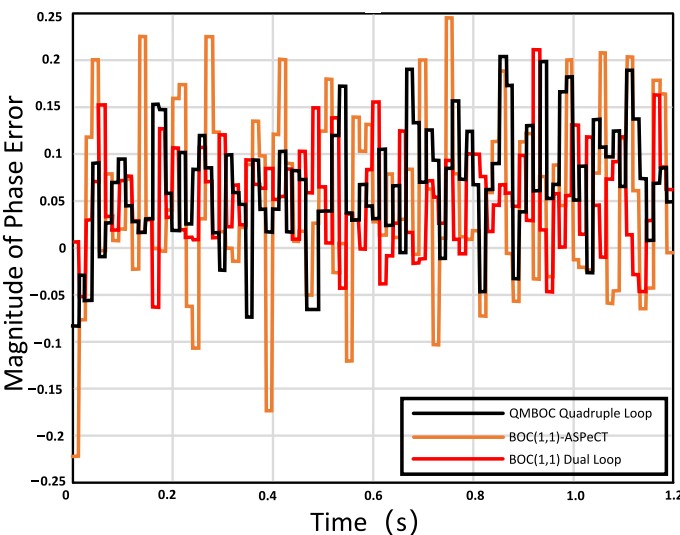

**Figure 17.** Pseudocode Loop Tracking Results.

In terms of pseudocode tracking results, both BOC(1,1)-ASPeCT and quadruple-loop tracking methods exhibit larger fluctuations in pseudocode tracking errors compared to BOC(1,1) pseudocode tracking. However, the dual-loop tracking method for BOC(1,1) demonstrates the smallest errors at this signal-to-noise ratio. In the context of signal algorithm simulation verification, the subcarrier component serves as an auxiliary component to achieve accurate tracking, with the ultimate goal of enhancing the accuracy and stability of signal tracking. From a stability analysis perspective, the signal-to-noise ratio has a significant impact on tracking algorithm research. The comparative chart of pseudocode tracking outputs under different signal-to-noise ratios is presented in Figure 18.

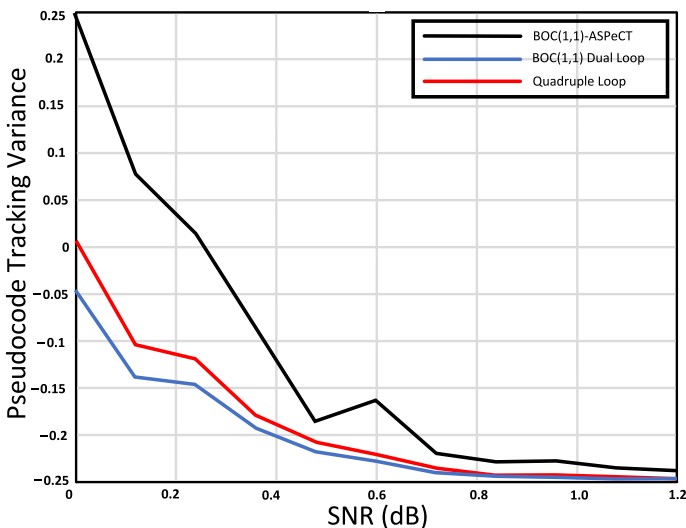

**Figure 18.** Comparison of Root Mean Square Error.

### 3.2. Practical Testing

Figure 19 illustrates the testing hardware platform designed for this study, which primarily consists of RF circuitry, baseband circuitry, clock circuitry, and interface circuitry. In the RF circuitry design, a dedicated RF chip is employed to replace the conventional discrete components. This integration allows for functions such as Low-Noise Amplifier (LNA), filtering, mixing, and analog Automatic Gain Control (AGC) to be performed by a single chip, greatly simplifying the design and debugging complexity of the RF section while ensuring consistency in receiver performance. In the baseband circuitry design, an FPGA chip based on the System On Programmable Chip (SOPC) architecture is adopted

as a replacement for the traditional FPGA+DSP approach. This design simplifies the circuitry complexity and reduces overall power consumption. The clock circuitry design utilizes a clock management chip with multiple channels, low jitter, and adjustable frequency and phase, providing coherent clocks to the RF section, Analog-to-Digital Converter (ADC) sampling, and FPGA, thereby ensuring that baseband performance is not compromised by clock signals. The power supply circuitry design employs a hybrid chip solution consisting of DC-DC and LDO regulators. LDO regulators are used for power-sensitive components requiring low ripple, such as the clock and RF sections, while DC-DC regulators are used for high power-demanding components like the baseband section, effectively balancing power quality and requirements. The interface section enables data transmission and control by communicating with a PC via Ethernet.

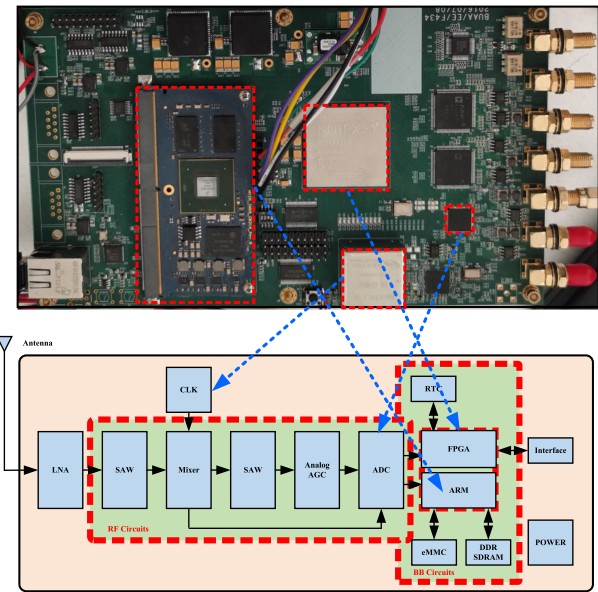

**Figure 19.** Testing Hardware Platform Design.

Figure 20 illustrates the actual test route for the tracking loop in this study. This route encompasses typical urban environments, including scenarios with building obstructions and signal reflections from structures. These conditions present challenges such as loss of lock, reacquisition, and tracking errors due to multipath interference. The test route encompasses three distinct challenging areas. One area is located near buildings and trees, where frequent obstructions and multipath effects occur. The other two areas involve traversing covered walkways within buildings, where the receiver does not have direct line-of-sight to the satellites.

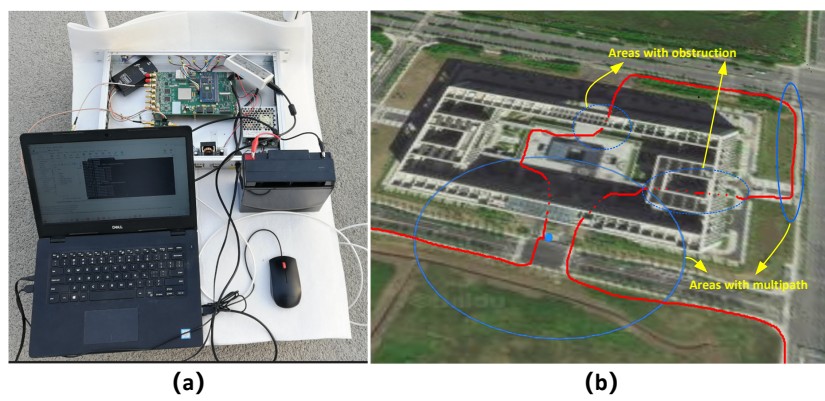

**Figure 20.** (**a**) Testing Hardware Platform. (**b**) Test Route.

Figure 20 illustrates the tracking status of the receiver's loop when passing through a large obstructed area. From Figure 21a, it can be observed that the tracking loop exhibits numerous "spikes" in the discriminator tracking error when encountering frequent obstructions. These spikes indicate brief moments of tracking loss. However, due to the predictive capability of the Kalman filter structure in the carrier loop, the loop quickly reacquires lock after the momentary loss. After an extended period of obstruction, the Kalman filter structure is unable to maintain a stable lock on the loop, resulting in complete loss of the lock. When satellite signals reappear, the loop reacquires the lock.

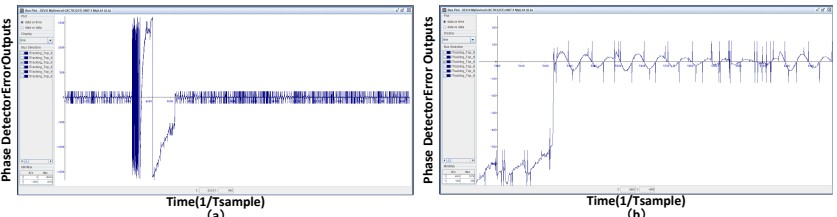

**Figure 21.** The tracking status of the loop when passing through obstructed areas. (**a**) Overview. (**b**) Details.

Figure 22a presents the instantaneous states of the FLL and PLL in the receiver's carrier loop before and after stable signal tracking. Specifically, Figure 22b shows the state of the FLL, while Figure 22c displays the state of the PLL. From Figure 22b, it can be observed that the FLL enters a stable state before the PLL. This is because during the tracking process, the FLL synchronizes the frequency components of the signal first, resulting in the local carrier generator's frequency matching the received signal's frequency (including Doppler). In Figure 22c, it can be observed that the PLL operates concurrently with the FLL, continuously synchronizing the phase of the received signal. As the FLL approaches the target frequency, the PLL's tracking gradually slows down. Once the phase of the target signal falls within the capture range of the PLL, it quickly acquires and tracks the signal, reducing the output error rapidly until it approaches zero. At this point, the loop has achieved a stable lock on the target signal.

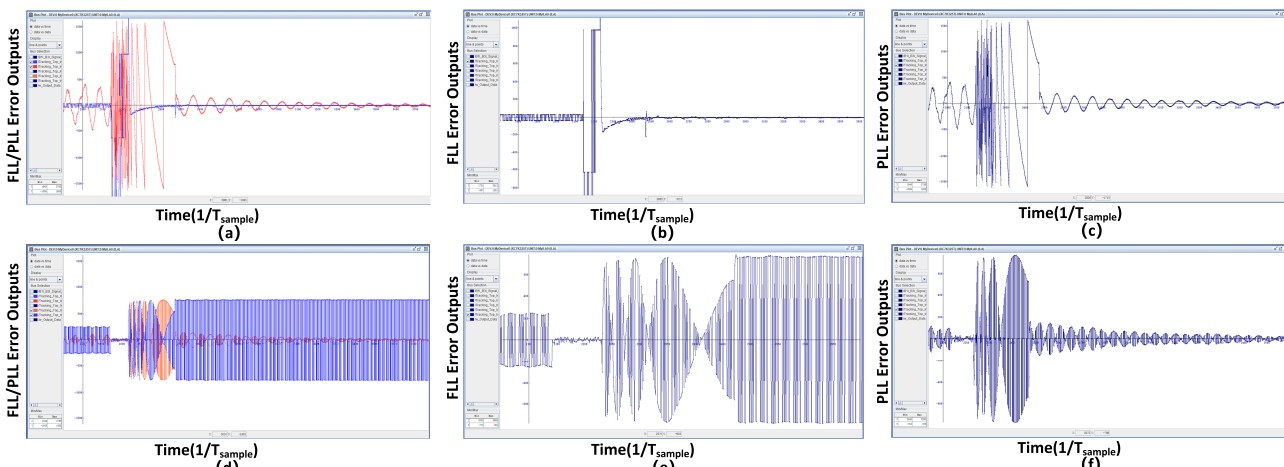

**Figure 22.** (**a**) The instantaneous states of the FLL and PLL in the receiver's carrier loop before and after stable signal tracking. (**b**) FLL Details. (**c**) PLL Details. (**d**) The instantaneous states of the prompt I branch and prompt Q branch in the receiver's tracking loop before and after achieving stability. (**e**) I branch Details. (**f**) Q branch Details.

The instantaneous states of the prompt in-phase (I) branch and prompt quadrature-phase (Q) branch in the receiver's tracking loop before and after achieving stability are depicted in Figure 22d. Figure 22e illustrates the transient state of the I branch, while Figure 22f presents the transient state of the Q branch. From Figure 22e, it can be observed

that the I branch reaches stability earlier than the Q branch. Prior to stable tracking, both branches exhibit fluctuations and transient behaviors. However, as the tracking loop approaches stability, the I branch demonstrates a more consistent and stable response compared to the Q branch. Figure 22f shows that the Q branch takes slightly longer to achieve stability. This delay can be attributed to the specific characteristics of the tracking algorithm and the signal being tracked. As the loop stabilizes, the Q branch aligns with the I branch, resulting in the accurate and synchronized tracking of the target signal.

## 4. Discussion

An urban environment is characterized by buildings, vehicles, and other structures that cause signal blockage and create multipath reflections. These factors lead to the loss of tracking loops and the degradation of signal quality. Additionally, the dynamic nature of urban environments introduces Doppler frequency shifts due to the movement of objects, such as vehicles or pedestrians.

Based on the simulation analysis conducted above, it can be concluded that the designed tracking loop for B1C signals effectively achieves synchronization with the simulated signals. Furthermore, a comparison of the results reveals the influence of environmental noise on the receiver's tracking loop and its corresponding performance. The analysis considers the multi-peak autocorrelation characteristics of B1C signals, the principles of the code phase discriminator, and the performance of the tracking loop. From this analysis, it is evident that the ambiguity of BOC signals is a significant factor contributing to the instability of the tracking loop.

The tracking errors decrease as the signal-to-noise ratio increases. The tracking results of the combined quadruple-loop tracking method are higher than those of the dual-loop tracking method for BOC(1,1), but they are significantly lower than the results obtained using the ASPeCT processing method. Furthermore, under conditions where the signal-to-noise ratio exceeds $-28$ dB, the tracking errors of the combined quadruple-loop tracking method remain within 0.03 chips. This suggests that within an appropriate signal-to-noise ratio range, the conditions for achieving high-precision signal tracking are met, enabling further enhancement of the tracking accuracy of the quadruple-loop tracking method.

The fundamental principle behind the code phase deblurring algorithm proposed in this paper is as follows: The code tracking loop generates a distinct but coarse delay estimate, while the sub-carrier tracking loop produces a higher-precision fine delay estimate, albeit inherently ambiguous. This algorithm can provide robust fault tolerance when significant distortion occurs due to large sampling errors and the influence of noise and multipath interference on pseudocode correlation. As the subcarrier correlation takes the form of a continuous triangular waveform with multiple peaks spaced at $1/2Ts$ intervals, stable tracking can be achieved in both dimensions, thereby demonstrating strong resistance to multipath and interference effects.

From the analysis of the data from the actual test path, Table 3 is derived, which compares the impact of the deblurring algorithm on receiver performance in different scenarios. It can be observed that, under consistent key parameters, the algorithm exhibits optimal performance in open areas. In multipath scenarios, there is some fluctuation in SNR due to complexity, which reduces the receiver's sensitivity. In the worst-case obstructed area, both receiver sensitivity and tracking error performance experience significant degradation. In this case, increasing the integration time can partially compensate for the performance degradation.

**Table 3.** The Impact of Test Route Conditions on Receiver Performance.

| Test Route Conditions | | | | | | | | | |
|---|---|---|---|---|---|---|---|---|---|
| **Visible Satellites Number** | **Code Correlation Interval (chip)** | **Subcarrier Correlation Interval (chip)** | **Loop Noise Bandwidth (Hz)** | **Carrier Loop Bandwidth (Hz)** | **SNR (dB)** | **Typical Scenarios** | **Average Tracking Time (ms)** | **Average Tracking Sensitivity (dBm)** | **Average Tracking Error (chip)** |
| ≥10 | 0.5 | 0.5 | 5 | 20 | ≥−20 | Open Environments | ≤50 | −153.72 | 0.03 |
| | | | | | −30~−20 | Multipath Environments | ≤50 | −150.32 | 0.25 |
| | | | | | ≤−40 | Obstructed Environments | ≤100 | −142.17 | 0.73 |

## 5. Conclusions

This paper addresses the numerous challenges faced by the application of BeiDou B1C signals in urban environments. Starting from the modeling of B1C signals in urban environments, a multi-loop structure suitable for synchronizing and tracking B1C signals is proposed. Based on this structure, a multi-peak phase deblurring algorithm specific to BeiDou B1C signals in urban environments is developed. This algorithm considers the coupling relationship between the code and carrier loops and achieves accurate and unambiguous phase estimation of the pseudocode by matching the structure of multiple loops, thus ensuring the stable tracking of BeiDou B1C signals. The performance of the algorithm is analyzed in terms of tracking accuracy, tracking sensitivity, and tracking time. Furthermore, the algorithm is implemented on a Software-Defined Radio (SDR) platform, and practical tests are conducted in typical challenging urban scenarios to validate the effectiveness of the algorithm. The algorithm exhibits an error of less than 0.03 for chip intervals when the signal-to-noise ratio exceeds $-20$ dB. Furthermore, precision enhancement can be achieved by modifying the set conditions, thereby fulfilling its applicability in urban environments. The signal tracking technology proposed in this paper effectively addresses the issues of tracking ambiguity and tracking accuracy for both components. Considering the characteristics of the dual-component signal structure, especially the absence of navigation messages on the pilot component, this opens up more possibilities and performance improvement potential for B1C signal tracking technology. In urban environments, the SNR often fluctuates, and for lower SNR conditions, the use of long-term integration on the pilot component can be considered to achieve higher tracking accuracy. Additionally, the design and implementation of alternative tracking algorithms can further enhance receiver performance.

**Author Contributions:** Conceptualization, X.Y. (Xu Yang 1) and C.Z.; methodology, W.F.; software, X.Y. (Xu Yang 1); validation, X.Y. (Xu Yang 1), Z.Y. and Q.W.; formal analysis, C.Z.; investigation, X.Y.; resources, X.Y. (Xu Yang 1); data curation, X.Y. (Xu Yang 1); writing—original draft preparation, X.Y. (Xu Yang 1); writing—review and editing, X.Y. (Xu Yang 1) and C.Z.; visualization, X.Y. (Xu Yang 2); supervision, C.Z.; project administration, W.F. All authors have read and agreed to the published version of the manuscript.

**Funding:** This research was funded by the National Natural Science Foundation of China under grant number 61901015.

**Institutional Review Board Statement:** Not applicable.

**Informed Consent Statement:** Not applicable.

**Data Availability Statement:** The data presented in this study are available on request from the corresponding author.

**Acknowledgments:** The authors acknowledge graduate student Xu Yang for his contribution to the literature search and collation.

**Conflicts of Interest:** The authors declare no conflict of interest.

## Abbreviations

The following abbreviations are used in this manuscript:

| | |
|---|---|
| ACF | Autocorrelation Function |
| ADC | Analog-to-Digital Converter |
| AGC | Automatic Gain Control |
| AltBOC | Alternative Binary Offset Carrier |
| ASPeCT | Adaptive Subspace Power Estimation and Cancellation Technique |
| BDS | BeiDou Navigation Satellite System |
| BOC | Binary Offset Carrier |
| BPSK | Binary Phase Shift Keying |
| BPSK-R | Binary Phase Shift Keying-Rectangular |

| | |
|---|---|
| CBOC | Composite Binary Offset Carrier |
| CAT | Cross-Assisted Tracking |
| CDMA | Code Division Multiple Access |
| CNR | Carrier to Noise Ratio |
| CWIs | Continuous Wave Interferences |
| DPE | Dual Phase Estimator |
| DLL | Delay Lock Loop |
| DRAM | Dynamic Random Access Memory |
| DSP | Digital Signal Processing |
| DSPs | Digital Signal Processors |
| DSSS-CDMA | Direct Sequence Spread Spectrum-Code Division Multiple Access |
| DualQPSK | Dual-Quadrature Phase Shift Keying |
| EKF | Extend Kalman Filter |
| ELS | Early Late Slope |
| EMLA | Early Minus Late Amplitude |
| EPL | Early-Prompt-Late |
| FDD | Frequency Division Duplexing |
| FIFO | First Input First Output |
| FIR | Finite Impulse Response |
| FLL | Frequency Locked Loop |
| FPGA | Field Programmable Gate Array |
| Galileo | Galileo Navigation Satellite System |
| GLONASS | GLONASS Navigation Satellite System |
| GNSS | Global Navigation Satellite System |
| GPS | Global Positioning System |
| GPU | Graphics Processing Unit |
| HRC | High Resolution Correlator |
| IF | Intermediate Frequency |
| IFFT | Inverse Fast Fourier Transformation |
| LNA | Low Noise Amplifier |
| LPF | Low Pass Filter |
| MCMM | Mixed Mode Clock Manager |
| MBOC | Multiplexed Binary Offset Carrier |
| NELP | Non-coherent Early-Late Power |
| PLL | Phase Lock Loop |
| PRN | Pseudorandom Noise |
| PSD | Power Spectral Density |
| QMBOC | Quadrature Multiplexed Binary Offset Carrier |
| SCBOC | Single-Sideband Complex Binary Offset Carrier |
| SCPC | Sub Carrier Phase Cancellation |
| SDR | Software-Defined Radio |
| SLL | Subcarrier Lock Loop |
| SOPC | System On Programmable Chip |
| TMBOC | Time-Multiplexed Binary Offset Carrier |

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
