# Peer review of "A Multi-Correlation Peak Phase Deblurring Algorithm for BeiDou B1C Signals in Urban Environments"

_remotesensing, doi:10.3390/rs15174300_

Round 1

Reviewer 1 Report

The paper introduces a phase deblurring algorithm to enhance the tracking on BeiDou B1C signals in degraded reception conditions (lower C/N0).

The introduction contains a thorough review of the literature but would benefit from being organised in sub-sections. It would also be beneficial to high-light the gaps in the different techniques presented, that are addressed by this research.

Figure 1 is introducing a good summary of the research but should be explained more in details. It would help the reader to link the different boxes to the sections of the paper where they are discussed.

In your first contribution (l.192), you mention that the mathematical model and structure of the BeiDou B1C signal is constructed based on the characteristics of urban environments. One would expect the signal model to be independent from the environment but to be necessary to then assess the impact of the reception conditions on the user performance. Please consider reformulating this statement.

The steps corresponding to the phase deblurring algorithm should be more clearly identified in section 2.2.2.

In section 3 (Results and Discussion), please make the figures' axis and legends larger. The code phase tracking errors observed for SNR = -20bB (Fig. 13) should be further discussed, in particular in terms of potential limitations for the receiver/user.

The discussion should be added at the end of the sub-section or in a separated sub-section (3.3), summarising the benefits and limitations of the new schemes when compared with the state-of-the art (maybe with a table?). A discussion on the impact of the multipath on the correlation function itself would have been expected.

In section 3.2, as the test route contains different reception conditions, a performance comparison between these conditions would be useful. Discussion on the results and their impact in terms of receiver performance should be included.

In the conclusions, the possible enhancements (l.717) should be further detailed. Given the topic of the special issue, it would also be benefic to link more clearly the work presented to the gain expected for urban transportation.

Typos/editorials:

- l.180 'the paper establishes a modeling and development framework for B1C signals in urban environments' -> it is not clear what is meant by a modelling and development framework

- l.224 'Furthermore, whether the spread spectrum signal modulates the data and the corresponding data rate also affect the ranging performance of the signal.' -> please reformulate

- Eq.1 BIC and B1C being used

- Fig. 2: indicate the difference between the 2 figures (BDS-2 and BDS-3?)

- l.625: the 40 times smaller error does not appear clearly on the figure.

Reviewer 2 Report

Dear Authors,

Manuscript is quite well written with respect to the experimentation. However, following modifications/corrections are suggested:

1. Note: "Open signals L1OC and L2OC use time-division multiplexing to transmit pilot and data signals, with BPSK(1) modulation for data and BOC(1,1) modulation for pilot". In this light kindly correct the line 39, as all four GNSS are using BOC or derivatives. Ofcourse Glonass use both FDMA and CDMA for compatibility reasons from 2006 onwards.

2. Summary part from line 191 onwards can be moved to results and discussion /conclusion part appropriately.

3. Figure 1 and associated text can be moved to Method/procedure section. introduction shall focus on Introduction with last para on the objective/aim of the paper.

Manuscript is overall well written which need few corrections. However, the distribution of content need to placed in correct section as per standard publication formats.

best wishes,

Statements/ sentences with - We / I /When ... in the beginning,  need to be reframed.
